# Exact Pose Actions versus Coordinate-Conditioned Rendering in Known-Pose Variational Autoencoders

## Abstract

Known-pose generative models often receive pose metadata such as object angle, camera state or acquisition geometry. We study a variational autoencoder (VAE) design choice: should the supplied pose be given to the decoder as coordinates or should the model decode canonical content and apply pose through a fixed geometric action? Within each experiment, the compared pathways use matched encoders, latent dimensions, decoder widths, losses, data splits, seeds and training budgets. We compare coordinate-conditioned rendering, a learned-warp control and exact-action pathways. The exact-action pathways first predict the object in a canonical orientation and then apply the known rotation with a fixed differentiable sampler. In the main two-dimensional silhouette experiment over ten paired seeds, the rotor-based exact-action pathway improves thresholded view Dice from $0.8406 \pm 0.0166$ to $0.9031 \pm 0.0189$ and reduces relative-pose composition mean-squared error from $0.0327 \pm 0.0029$ to $0.0104 \pm 0.0007$, while the coordinate decoder obtains lower probabilistic view binary cross-entropy and better canonical binary cross-entropy/Dice. Scaling, procedural polygons, rotated handwritten digits, loss ablations and a controlled three-dimensional voxel study show the same composition consistency pattern, although the exact pathway does not win every reconstruction metric. Geometric Algebra is used to express exact-actions and their composition, not to claim that a planar rotor is a better implementation than an equivalent rotation-matrix or restricted spatial-transformer warp. In two dimensions these are the same action. In the three-dimensional rotation only study spatial rotors/unit quaternions induce the matrices consumed by the voxel sampler, while full rigid motion motors are left for future work. The empirical claim is architectural, asserting that when pose is known and appearance changes are geometric, applying pose as a fixed action can improve shape fidelity and relative-pose consistency in controlled variational autoencoders.

## 1 Introduction

Pose metadata are available in many generative modelling settings, including synthetic rendering pipelines, robotic sensing, calibrated multi-view object data and scientific acquisition protocols such as microscope stage rotation, tomographic acquisition and calibrated laboratory imaging. In these settings the main difficulty is not always to infer pose from the image, but instead the model must decide how to use a pose that is already supplied. A common neural choice is to concatenate pose coordinates to the decoder and let training discover how those coordinates alter the output, whereas an exact-action choice is to decode canonical content and apply the supplied transformation through a fixed geometric operation.

This distinction matters because the two pathways allocate learning capacity differently. If the pose input is in coordinates, the decoder must learn object content together with the transformation rule, while if pose is applied analytically the decoder models canonical content and the externally supplied transformation is handled by a fixed differentiable operator. Spatial transformer and geometry-aware generative models show the value of differentiable warping and explicit geometry (Jaderberg et al., 2015; Skafte & Hauberg, 2019; Kosiorek et al., 2021; Chan et al., 2022), yet the specific known-pose question is often entangled with pose estimation, photorealistic rendering, camera calibration or dataset complexity. The gap addressed here is therefore narrower and directly testable. When pose is already known, should the model relearn its action

from coordinates, or should that action be imposed exactly? For example if a silhouette is observed at $30°$, a coordinate-conditioned decoder is asked to map the shared content code together with $[\cos 30°, \sin 30°]$ directly to the rotated pixels. An exact-action decoder instead predicts the canonical silhouette once and applies the known $30°$ rotation with a fixed sampler.

> **Research question.** In a known-pose paired view VAE, with architecture, optimiser, loss weights, training schedule, seed set and data split held fixed, does routing the supplied pose through an exact analytic action improve posed-view fidelity and relative-pose consistency compared with giving the same pose to a coordinate-conditioned decoder?

We answer this question with a controlled study. The main task uses binary asymmetric silhouettes because canonical targets, deterministic posed-views and ground-truth angles can be generated procedurally. This makes it possible to match all non-pose choices across models, including optimiser, width, training schedule, loss weights and data split. The setting is a controlled test of a pose pathway rather than a natural image benchmark, as it isolates whether the pathway has the intended inductive bias before adding texture, illumination, object detection, camera estimation or learned photorealistic rendering. Building on this test, we add ten-seed paired statistics, a learned-warp control, an exact matrix spatial transformer control restricted to rotation, data and capacity scaling, angle generalisation tests, procedural shape tests with random asymmetric polygons, Rotated MNIST, loss ablations and a controlled $3D$ voxel experiment.

Geometric Algebra (GA) provides a compact notation for exact geometric actions. In the $2D$ Clifford algebra $\text{Cl}(2,0)$, let $e_1$ and $e_2$ be orthonormal basis vectors and let $\mathbf{I} = e_1 e_2$ be the unit pseudoscalar which satisfies $\mathbf{I}^2 = -1$. The planar rotor $r(\theta) = \cos(\theta/2) - \sin(\theta/2)\mathbf{I}$ induces the same Euclidean rotation through angle $\theta$ as the standard SO(2) matrix and the implementation ultimately uses the same inverse sampling operation as a rotation only spatial transformer. Thus, the $2D$ result is not a claim that GA is empirically superior to matrices. It is a matched comparison between exact known-pose action and learned coordinate-conditioned rendering. We keep the GA language because the same sandwich action notation extends from planar rotors to spatial rotors for $3D$ rotations and to motors for rigid motions that include translations (Doran & Lasenby, 2003; Lasenby et al., 2024; Matsantonis, 2025). In the $3D$ rotation only experiment, rotations are sampled as unit quaternions which are coordinate representations of spatial rotors in the even subalgebra of $\text{Cl}(3,0)$ up to basis convention. The induced SO(3) matrices are stored because the differentiable voxel sampler consumes affine coordinate maps. We therefore test a single exact SO(3) action rather than a separate $3D$ rotor versus matrix contest. The appendices give both the algebraic derivation and an independent `clifford` library (Hadfield et al., 2021) verification of the $2D$ and $3D$ rotor to matrix conversions. We do not use conformal or projective motors in this paper because the tested $3D$ task contains rotations about the voxel grid centre, not full SE(3) rigid motions with translations.

The contribution is threefold. First, we formulate a falsifiable known-pose VAE comparison in which the intended structural difference is the pathway by which pose enters generation. Second, we show empirically that exact analytic actions improve thresholded posed-view Dice and composition consistency in the controlled $2D$ setting, while reporting the coordinate decoder's lower probabilistic view cross-entropy. Third, we use scaling, ablations, learned-warp controls, Rotated MNIST and a $3D$ voxel extension to identify the boundary of the claim. The evidence supports an exact-action and sample efficiency interpretation, not a universal reconstruction superiority claim.

## 2 Related work

Variational Autoencoders (VAEs) provide the latent variable framework used in this paper (Kingma & Welling, 2014). In light of this, disentanglement methods such as $\beta$-VAE show that regularisation can encourage factor separation (Higgins et al., 2017), but our setting differs because pose is supplied and the question is how to route it through the decoder. Moreover, transforming autoencoders and spatial transformer networks introduced transformation-aware representations and differentiable warping modules (Hinton et al., 2011; Jaderberg et al., 2015). The Variationally Inferred Transformational Autoencoder (VITAE) then combined VAEs with spatial transformers to separate appearance and perspective (Skafte & Hauberg, 2019). Our

comparison is narrower than these systems because it does not learn pose from the image. Instead, it asks whether a known pose should be implemented as a fixed action or consumed by a learned decoder.

Group equivariant architectures impose transformation structure throughout a network (Cohen & Welling, 2016) and geometry-aware generative models often include explicit $3D$ structure or camera-conditioned rendering (Kosiorek et al., 2021; Chan et al., 2022). Those works motivate the broad importance of geometric structure, whereas our study isolates a smaller architectural choice inside a known-pose VAE. GA neural networks and Clifford-algebra architectures use multivectors and algebraic group actions inside neural modules (Ruhe et al., 2023b;a). Our model is not a full multivector-valued neural network. It inserts one fixed analytic action into the decoder and compares it against alternatives with the same encoder, capacity, losses, data and optimisation budget.

The paired view encoder also relates to multimodal VAEs. That said, mixture of experts and generalized multimodal evidence lower bound (ELBO) formulations provide principled ways to combine evidence from multiple modalities or views (Shi et al., 2019; Sutter et al., 2021). We do not claim a new multimodal variational bound. The same shared latent construction is used for all models so that differences in performance can be attributed to the pose pathway rather than to a different inference objective, for example a mixture of experts or product of experts objective.

## 3 Controlled Problem and Model Family

Each $2D$ example contains a canonical binary silhouette $c \in \{0,1\}^{H \times W}$, where $H$ and $W$ are the image height and width. It also contains two known image space angles $\theta_a, \theta_b \in [-\pi, \pi)$ and two observed silhouettes $x_a, x_b$ obtained by rotating $c$ on the raster grid. The subscripts $a$ and $b$ denote the two views of the same object. The latent content variable is $z \in \mathbb{R}^d$, where $d$ is the latent dimension. For view $x_v$ with $v \in \{a, b\}$, the encoder returns a diagonal Gaussian content posterior:

$$q_\omega(z \mid x_v) = \mathcal{N}(\mu_v, \text{diag}(\sigma_v^2)) \tag{1}$$

where $\omega$ denotes the encoder parameters, $\mu_v \in \mathbb{R}^d$ is the posterior mean and $\sigma_v^2 \in \mathbb{R}_+^d$ is the diagonal variance. A reparameterised sample from this posterior is denoted $z_v$. For a fair comparison, every model forms the same shared content sample and deterministic shared mean:

$$z_\text{s} = \frac{1}{2}(z_a + z_b) \qquad \mu_\text{s} = \frac{1}{2}(\mu_a + \mu_b) \tag{2}$$

This averaging step is a controlled fusion device used by all compared pathways. It is not proposed as a new multimodal evidence lower bound and Appendix F gives the complete architectural details.

The analytic model first decodes the shared content into a canonical logit field and then applies the supplied pose with a fixed sampling operator. Here a logit field means the unnormalised occupancy scores before the sigmoid nonlinearity used for Bernoulli probabilities. Let $D_\psi$ denote the canonical decoder with parameters $\psi$ and let $\mathcal{W}_\theta$ denote the inverse sampling image warp for angle $\theta$. The deterministic mean field path used for thresholded Dice, canonical metrics and composition evaluation is:

$$\bar{\ell}_\text{can} = D_\psi(\mu_\text{s}) \qquad \bar{\ell}_v = \mathcal{W}_{\theta_v} \bar{\ell}_\text{can} \qquad v \in \{a, b\} \tag{3}$$

The coordinate-conditioned decoder instead receives pose as ordinary coordinates and directly predicts each posed-view:

$$\bar{\ell}_v^\text{coord} = D_\psi([\mu_\text{s}, \cos\theta_v, \sin\theta_v]) \qquad v \in \{a, b\} \tag{4}$$

The analytic pathway therefore learns a canonical field and uses the known transformation to place it in each view. The coordinate pathway learns how changing $[\cos\theta, \sin\theta]$ changes pixels, because no fixed geometric relation is imposed after decoding. During training both pathways use the same sampled view binary cross-entropy (BCE) reconstruction terms, sampled view Dice term, deterministic canonical supervision terms, content invariance term, Kullback-Leibler (KL) regularisation term and confidence term:

$$\begin{aligned} \mathcal{L} = \mathcal{L}_\text{view-BCE} + \lambda_\text{view-Dice}\mathcal{L}_\text{view-Dice} + \lambda_\text{can-BCE}\mathcal{L}_\text{can-BCE} \\ + \lambda_\text{can-Dice}\mathcal{L}_\text{can-Dice} + \lambda_\text{inv}\frac{1}{d}\|\mu_a - \mu_b\|_2^2 + \beta_z\mathcal{L}_\text{KL} + \lambda_\text{conf}\mathcal{L}_\text{conf} \end{aligned} \tag{5}$$

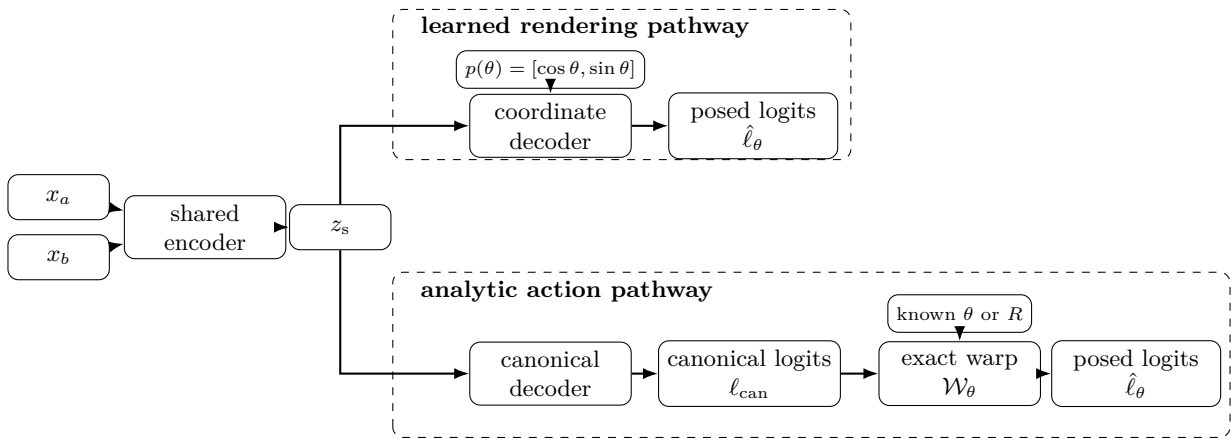

Figure 1: The controlled comparison isolates where pose enters generation. The learned rendering pathway gives the decoder content and pose coordinates and asks it to produce posed logits directly. The analytic action pathway decodes canonical logits once and applies the supplied pose through a fixed differentiable warp. Here logits are unnormalised occupancy scores before the sigmoid.

Here $d$ is the latent dimension. Appendix E defines every term in the implemented training objective, including which logits are sampled latent logits and which are deterministic mean path logits. Canonical supervision is included because the canonical output is evaluated directly. It also gives the coordinate decoder a fair canonical target, i.e. its canonical prediction is obtained by setting the pose code to zero, but posed-view reconstruction only trains the decoder at the two observed angles, so without a canonical term the zero-angle output would be weakly constrained. Appendix D explains this design choice and the use of asymmetric shapes.

Figure 1 shows the two high-level routes by which pose enters generation. Within these routes, we compare four $2D$ pathways. The coordinate decoder receives $[z_\text{s}, \cos\theta, \sin\theta]$ and directly outputs logits for the posed-view. The learned-warp control decodes canonical logits but uses a small pose network to predict a rotation for the sampler which tests whether a spatial sampler alone explains the effect. The exact $\mathrm{SO}(2)$/spatial transformer network (STN) row decodes canonical logits and applies the known rotation matrix with bilinear inverse sampling. The GA rotor row first uses the rotor coefficients $\cos(\theta/2)$ and $\sin(\theta/2)$ to construct the same rotation matrix, then calls the same sampler. *Therefore, exact $\mathrm{SO}(2)$/STN and GA rotor should agree up to numerical and interpolation details.* This agreement is a consequence of the isomorphism between planar unit rotors and planar rotations and it is used as a transparency check rather than as a claim of GA superiority. In $3D$, we report one exact-action row rather than separate matrix and rotor rows, because a unit spatial rotor, a unit quaternion and its induced $\mathrm{SO}(3)$ matrix encode the same known rotation for the rotation only voxel task, and the sampler ultimately requires the induced matrix.

The $2D$ analytic action is defined as follows. Let $R(\theta) \in \mathrm{SO}(2)$ denote the active Euclidean rotation induced by either a matrix or a rotor. The decoder predicts canonical logits $\ell_\text{can}$ and the posed logit field is obtained by inverse sampling:

$$\ell_\theta(u) = \ell_\text{can}\big(R(\theta)^{-1}u\big) \tag{6}$$

where $u$ is a normalised image grid coordinate. Because the raster vertical axis points downward, a positive image space rotation uses the Euclidean rotor for $-\theta$, and Appendix A gives the derivation and sign convention.

## 4 Experimental Protocol

Table 1 summarises the datasets, sizes and optimisation settings used throughout the paper, and Appendix F gives the corresponding architectural details. The main $2D$ silhouettes are drawn from six hand designed asymmetric templates because the first experiment is intended to isolate the pose pathway rather than test content generalisation. A single template would make content nearly trivial, whereas a large random

shape distribution would entangle pose handling with a harder shape modelling problem. Six templates give repeated but non-identical canonical objects, exact canonical targets and no obvious non-trivial rotational symmetries, so the supplied angle has an identifiable meaning. To check that the result is not merely a fixed template artefact, we also include angle generalisation, procedural asymmetric polygon and Rotated MNIST known-pose checks (their results are reported in Section 5).

In the main $2D$ setting, the number of training pairs is $N = 256$, the number of validation pairs is 128, the image size is $32 \times 32$, the latent dimension is $d = 16$, the decoder base width is $C = 8$, and the batch size is 32. We optimise with Adam at learning rate $2 \times 10^{-3}$ (Kingma & Ba, 2015). The fixed $2D$ budget is 32 epochs, which corresponds to 256 optimiser updates in the main 256-pair setting. This compressed regime is used because the research question concerns sample efficiency and structural consistency when capacity is constrained. To test whether the conclusion depends only on that regime, we also run a scaling grid with $N \in \{256, 1024, 4096\}$ training pairs and capacities $(d, C) \in \{(16, 8), (32, 16), (64, 32)\}$. The largest grid point has 16 times more training pairs than the main setting. We therefore treat the grid as a finite scaling test. It shows how the effect changes over this measured range without claiming what would happen at still larger datasets or wider networks.

The scaling grid uses independently generated train/validation splits for each grid cell, so the $256/(16, 8)$ scaling row is not expected to be numerically identical to the main $2D$ table.

Table 1: Experimental matrix. The same optimiser and loss weights are shared by all compared models within a given experiment. Only the pose pathway changes unless the row explicitly describes a data or capacity sweep.

| Experiment | Data and split | Capacity and optimiser | Repetitions |
|---|---|---|---|
| Main $2D$ silhouettes | $32 \times 32$ binary pairs, 256 train, 128 validation | $d = 16$, $C = 8$, batch 32, Adam $2 \times 10^{-3}$, 32 epochs | 10 paired seeds |
| Scaling grid | $N \in \{256, 1024, 4096\}$ train, validation $\min(\max(128, N/4), 1024)$ | $(d, C) \in \{(16, 8), (32, 16), (64, 32)\}$, same optimiser, 32 epochs | 10 paired seeds |
| Angle and procedural shapes | 512 train, 256 validation, restricted-angle or random polygon data | $d = 16$, $C = 8$, same optimiser, 32 epochs | 10 paired seeds |
| Loss ablations | Main $2D$ data and split | $d = 16$, $C = 8$, same optimiser, 32 epochs, one loss component changed per row | 10 paired seeds |
| Rotated MNIST | 12,000 train, 2,000 validation, thresholded and padded digits | $d = 32$, $C = 16$, batch 256, Adam $2 \times 10^{-3}$, 32 epochs | 10 paired seeds |
| $3D$ voxels | $32^3$ binary volumes, 512 train, 128 validation | $d = 32$, $C = 8$, batch 8, Adam $10^{-3}$, 24 epochs | 10 paired seeds |

The rationale for the compared pathways and datasets is summarised in Appendix G. In brief, the learned-warp baseline separates the effect of using a sampler from the effect of applying the supplied action exactly, the exact SO(2)/STN control verifies that the $2D$ claim is not GA-specific, and the procedural polygon, Rotated MNIST and $3D$ voxel settings test the scope of the effect beyond six fixed silhouettes.

All comparisons within each experiment share the same data split, seed, optimiser, loss weights, batch size, and capacity. We do not perform per-model hyperparameter optimisation with tools such as Optuna (Akiba et al., 2019). This is a design choice rather than a claim that the fixed settings are optimal. The goal is to isolate the effect of the pose pathway, and a separate learning rate, width, batch size, or loss weight search for each pathway would make the performance gap partly attributable to tuning effort. The main comparisons use ten paired seeds, 0 through 9, which provides a direct seed level pairing for the four pathways. Paired confidence intervals and paired $t$-tests are reported in Appendix H and Table A6.

We report Bernoulli binary cross-entropy (BCE), thresholded Dice overlap and relative-pose composition mean-squared error (MSE). The table column View BCE is the validation sampled latent two-view reconstruction BCE sum, $\text{BCE}(\ell_a^z, x_a) + \text{BCE}(\ell_b^z, x_b)$ and is therefore treated as a probabilistic reconstruction diagnostic. Thresholded Dice, canonical metrics and composition MSE use deterministic mean path logits. BCE is the Bernoulli negative log-likelihood used in the VAE reconstruction term, while Dice is a standard binary overlap metric for segmentation and occupancy evaluation (Milletari et al., 2016). The composition metric analytically rotates predicted view $a$ by the known relative pose and compares it with predicted view $b$, where

views $a$ and $b$ are the two supplied observations of the same object. This metric checks whether learned outputs respect the supplied group action. It is not an independent benchmark of natural-image quality. Full formulas appear in Appendix E, and compact per-seed values for the key view Dice and composition metrics appear in Tables A3, A4 and A5.

The Rotated MNIST and $3D$ voxel checks follow the same fixed known-pose protocol. Their data sizes and optimisation settings are in Table 1, while preprocessing, sampler conventions and $3D$ rotation parameterisation are detailed in Appendix F and Appendix B. In brief, Rotated MNIST replaces polygon templates with real digit shapes under synthetic known rotations and the $3D$ voxel task replaces planar angles with sampled SO(3) rotations about the volume centre. These checks are robustness tests, not state of the art MNIST or photorealistic $3D$ rendering benchmarks.

## 5 Results

### 5.1 Exact known-pose actions improve view Dice and composition consistency in $2D$

Table 2 supports the central controlled claim while also showing its boundary after the longer 32-epoch training budget. The exact SO(2)/STN and GA rotor rows are close on the reported metrics which is the expected outcome because they implement the same $2D$ analytic action through the same sampler. Relative to the coordinate decoder, the GA rotor improves view Dice from $0.8406 \pm 0.0166$ to $0.9031 \pm 0.0189$ with a paired mean difference of 0.0625 and a 95% confidence interval of $[0.0484, 0.0767]$ in Appendix H. It also reduces composition MSE from $0.0327 \pm 0.0029$ to $0.0104 \pm 0.0007$ with a paired oriented difference of 0.0222 and a 95% confidence interval of $[0.0202, 0.0242]$. These two effects are the strongest evidence for the exact-action pathway.

Table 2: Main $2D$ comparison over ten paired seeds. Values are mean $\pm$ standard deviation. Best values are bold. Lower is better for BCE and MSE and higher is better for Dice. View BCE is the sampled latent two-view validation BCE sum; thresholded Dice, canonical metrics and composition MSE use deterministic mean path logits.

| Model | Canonical BCE ↓ | Canonical Dice ↑ | View BCE ↓ | View Dice ↑ | Composition MSE ↓ |
|---|---|---|---|---|---|
| Coord. decoder | **0.0356 ± 0.0122** | **0.9358 ± 0.0297** | **0.2165 ± 0.0166** | 0.8406 ± 0.0166 | 0.0327 ± 0.0029 |
| Learned warp | 0.0554 ± 0.0156 | 0.8932 ± 0.0388 | 0.2855 ± 0.0278 | 0.7811 ± 0.0192 | 0.0436 ± 0.0095 |
| Exact SO(2)/STN | 0.0494 ± 0.0090 | 0.9114 ± 0.0146 | 0.2486 ± 0.0173 | 0.9003 ± 0.0133 | **0.0103 ± 0.0006** |
| GA rotor | 0.0480 ± 0.0110 | 0.9148 ± 0.0221 | 0.2460 ± 0.0207 | **0.9031 ± 0.0189** | 0.0104 ± 0.0007 |

The same table prevents an overbroad interpretation. The coordinate decoder obtains the best sampled-latent two-view View BCE, $0.2165 \pm 0.0166$, whereas the GA rotor obtains $0.2460 \pm 0.0207$. This is consistent with a probabilistic reconstruction trade-off in which a flexible coordinate decoder can lower the sampled reconstruction BCE by producing smoother boundary probabilities, while an exactly warped binary-like canonical field can incur larger BCE under small raster misalignments. With the longer budget, the coordinate decoder also has the best canonical BCE and canonical Dice. We therefore treat View BCE as a reconstruction diagnostic and do not claim a canonical reconstruction advantage for the analytic pathway (the supported positive claim is about thresholded posed-view fidelity and relative-pose structure).

Appendix H reports per-seed values and paired tests, while Appendix I contains qualitative examples. The learned-warp control is informative because it has a spatial sampler but does not receive the exact known action. Its view Dice and composition MSE remain close to the coordinate decoder rather than the exact-action rows. Thus, in this controlled setting, the gain is not explained merely by the presence of a differentiable warp layer. The exact use of the supplied pose is the important structural difference.

### 5.2 Scaling, angle generalisation and procedural shapes test the scope of the effect

Table 3 shows why the claim is phrased as an exact-action and sample efficiency claim rather than as universal reconstruction superiority. The scaling grid was launched with ten paired seeds per cell. Because the grid

uses the same fixed optimiser, learning rate, loss weights and epoch budget across all capacities, a small number of high capacity runs produced non-finite validation quantities. We therefore report each row using only valid paired comparisons, where a pair is valid only if both the coordinate decoder and the GA rotor have finite view Dice and composition MSE values for the same seed. This paired filtering prevents one-sided non-finite runs from distorting the comparison. Most cells use all ten seeds, the two rows with $n_{\text{valid}} = 9$ exclude one non-finite paired comparison each. The largest $4096/(64, 32)$ cell produced only three valid paired comparisons, so we omit it from the main table rather than interpreting an underpowered and potentially selection-biased mean.

Table 3: Data and capacity scaling reported as raw paired means rather than relative percentages. The planned grid used ten paired seeds per cell. The column $n_{\text{valid}}$ is the number of seed pairs for which both pathways produced finite view Dice and composition MSE values, means and standard deviations in that row are computed over those valid paired seeds only. The table displays cells with at least nine valid paired seeds. The omitted $4096/(64, 32)$ cell had only three valid paired seeds, so we do not use it to support the scaling claim.

| Data $N$ / capacity $(d, C)$ | $n_{\text{valid}}$ | Coord. Dice | GA Dice | Coord. Comp. | GA Comp. |
|---|---|---|---|---|---|
| 256 / (16,8) | 10 | 0.838±0.012 | **0.877±0.041** | 0.0311±0.0018 | **0.0103±0.0007** |
| 256 / (32,16) | 10 | 0.872±0.016 | **0.918±0.045** | 0.0320±0.0014 | **0.0109±0.0008** |
| 256 / (64,32) | 10 | 0.877±0.023 | **0.912±0.069** | 0.0323±0.0029 | **0.0106±0.0013** |
| 1024 / (16,8) | 10 | 0.942±0.009 | **0.981±0.012** | 0.0285±0.0016 | **0.0124±0.0006** |
| 1024 / (32,16) | 10 | 0.966±0.006 | **0.990±0.015** | 0.0268±0.0010 | **0.0123±0.0006** |
| 1024 / (64,32) | 9 | 0.968±0.009 | **0.990±0.011** | 0.0276±0.0008 | **0.0127±0.0004** |
| 4096 / (16,8) | 10 | **0.977±0.009** | 0.976±0.032 | 0.0268±0.0005 | **0.0127±0.0011** |
| 4096 / (32,16) | 9 | 0.967±0.057 | **0.979±0.022** | 0.0270±0.0003 | **0.0125±0.0007** |

The GA rotor is ahead on view Dice at $N = 256$ and $N = 1024$ while the gap becomes comparable at $N = 4096$. For example at $4096/(16, 8)$ the coordinate decoder is $0.977 \pm 0.009$ and the GA rotor is $0.976 \pm 0.032$. In contrast, the composition MSE of the GA rotor remains lower, typically around 0.010-0.013 versus 0.027-0.032 for the coordinate decoder. This pattern suggests that a sufficiently trained coordinate decoder can approach the analytic pathway on thresholded reconstruction, while the exact-action continues to impose a more consistent relative-pose structure in the tested regime. The scaling table should therefore be read as a finite grid robustness check under shared hyperparameters, not as a fully tuned scaling law.

Table 4: Angle generalisation and procedural shape robustness over ten seeds. Best values are bold within each setting. The first setting trains only on angles with absolute value at most 60° and tests on uniform angles. The second trains on multiples of 45° and tests on uniform angles. The third replaces the six fixed templates by random asymmetric polygons.

| Setting | Model | View Dice ↑ | Composition MSE ↓ |
|---|---|---|---|
| range60 to uniform | Coord. decoder | 0.8367 ± 0.0122 | 0.0458 ± 0.0029 |
| | Learned warp | 0.7783 ± 0.0115 | 0.0559 ± 0.0039 |
| | Exact SO(2)/STN | **0.9371 ± 0.0190** | 0.0122 ± 0.0006 |
| | GA rotor | 0.9141 ± 0.0375 | **0.0121 ± 0.0008** |
| sparse45 to uniform | Coord. decoder | 0.8903 ± 0.0178 | 0.0337 ± 0.0016 |
| | Learned warp | 0.7806 ± 0.0229 | 0.0617 ± 0.0064 |
| | Exact SO(2)/STN | 0.9391 ± 0.0427 | **0.0125 ± 0.0014** |
| | GA rotor | **0.9402 ± 0.0396** | 0.0128 ± 0.0009 |
| procedural shapes | Coord. decoder | 0.7341 ± 0.0135 | 0.0245 ± 0.0008 |
| | Learned warp | 0.7254 ± 0.0135 | 0.0257 ± 0.0013 |
| | Exact SO(2)/STN | **0.7347 ± 0.0147** | 0.0101 ± 0.0007 |
| | GA rotor | 0.7338 ± 0.0143 | **0.0100 ± 0.0007** |

Table 4 separates two forms of robustness. Under restricted angle training, the exact-action rows preserve higher view Dice and much lower composition MSE on uniformly sampled test angles which is expected

because the supplied pose is applied by a fixed transformation rather than inferred as a coordinate to image rule. On procedural asymmetric polygons, the view Dice differences are essentially tied with the coordinate decoder at $0.7341 \pm 0.0135$, the exact SO(2)/STN row at $0.7347 \pm 0.0147$ and the GA rotor at $0.7338 \pm 0.0143$. However, composition MSE remains much lower for the exact rows, about 0.010 versus 0.0245 for the coordinate decoder. This suggests that the composition consistency result is not specific to the six hand designed templates, while the experiment remains a controlled binary shape test rather than a natural image benchmark.

**Rotated MNIST check.** Table 5 and Figure 2 report the Rotated MNIST experiment described in Section 4. The canonical target is the original upright digit and the two observed views are synthetically rotated versions of the same digit (LeCun et al., 1998). This setting preserves known pose and canonical supervision while replacing polygon templates by real digit shapes.

Table 5: Rotated MNIST known-pose robustness check over ten paired seeds. Best values are bold. View BCE is the sampled latent two-view validation BCE sum, thresholded Dice, canonical metrics and composition MSE use deterministic mean path logits. The task uses real digit shapes with synthetic known rotations, so it tests whether the same pathway pattern survives beyond polygonal silhouettes.

| Model | Canonical BCE ↓ | Canonical Dice ↑ | View BCE ↓ | View Dice ↑ | Composition MSE ↓ |
|---|---|---|---|---|---|
| Coord. decoder | $0.1520 \pm 0.0317$ | $0.7470 \pm 0.0660$ | $\mathbf{0.3647 \pm 0.0648}$ | $0.7602 \pm 0.0513$ | $0.0297 \pm 0.0020$ |
| Learned warp | $0.1358 \pm 0.0150$ | $0.7751 \pm 0.0321$ | $0.4944 \pm 0.0064$ | $0.5084 \pm 0.0093$ | $0.0951 \pm 0.0107$ |
| Exact SO(2)/STN | $0.1363 \pm 0.0097$ | $0.7825 \pm 0.0206$ | $0.4238 \pm 0.0223$ | $0.7892 \pm 0.0175$ | $\mathbf{0.0107 \pm 0.0001}$ |
| GA rotor | $\mathbf{0.1353 \pm 0.0049}$ | $\mathbf{0.7863 \pm 0.0103}$ | $0.4204 \pm 0.0109$ | $\mathbf{0.7940 \pm 0.0077}$ | $\mathbf{0.0107 \pm 0.0001}$ |

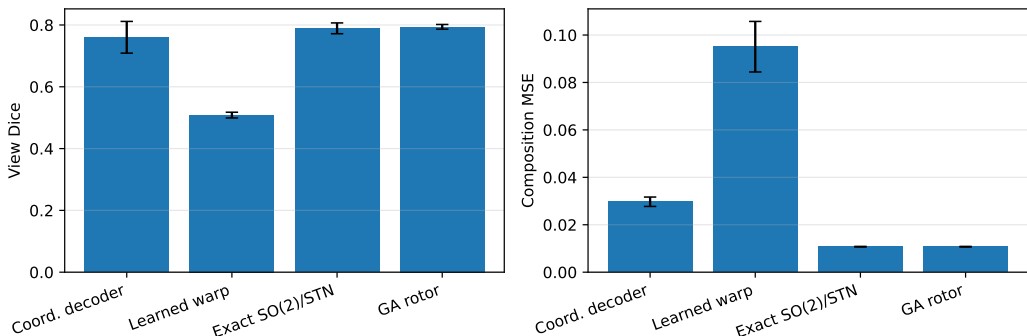

Figure 2: Rotated MNIST summary. The exact-action rows improve view Dice and reduce composition MSE relative to the coordinate decoder, while the coordinate decoder keeps the best sampled latent View BCE diagnostic as shown in Table 5.

The Rotated MNIST reconstruction picture is more mixed than the polygonal task, which is expected because digits introduce real shape variability and ambiguous canonical stroke patterns. The GA rotor improves view Dice from $0.7602 \pm 0.0513$ to $0.7940 \pm 0.0077$ and reduces composition MSE from $0.0297 \pm 0.0020$ to $0.0107 \pm 0.0001$. The coordinate decoder again has the lowest view BCE. The learned-warp control performs poorly on view Dice and composition in this setting, which reinforces the distinction between having a sampler and applying the supplied pose exactly. Appendix H reports compact per-seed Rotated MNIST values and Appendix I includes the qualitative Rotated MNIST panel.

### 5.3 Loss ablations show that the result is not driven by one hand-set term

Table 6 addresses the possibility that the thresholded metrics are driven by one hand-set loss term. Each row is a loss configuration ablation rather than a new model architecture, i.e. for each variant both the coordinate decoder and the GA rotor are retrained with the same architecture, data split, seed set, optimiser and epoch budget, while only the listed loss weights are changed. The variant names in the first column

Table 6: Targeted loss configuration ablations over ten seeds. Each row changes only the loss weights named in the `Loss variant` column. Both decoder types are retrained under that same configuration. Best values are bold within each loss variant.

| Loss variant | Model | View Dice ↑ | Comp. MSE ↓ | Entropy ↓ |
|---|---|---|---|---|
| `full` | Coord. decoder | $0.8403 \pm 0.0128$ | $0.0328 \pm 0.0024$ | $\mathbf{0.0418 \pm 0.0052}$ |
| | GA rotor | $\mathbf{0.8995 \pm 0.0204}$ | $\mathbf{0.0100 \pm 0.0008}$ | $0.0844 \pm 0.0087$ |
| `no_conf` | Coord. decoder | $0.8396 \pm 0.0143$ | $0.0328 \pm 0.0019$ | $\mathbf{0.0434 \pm 0.0066}$ |
| | GA rotor | $\mathbf{0.9033 \pm 0.0316}$ | $\mathbf{0.0101 \pm 0.0009}$ | $0.0842 \pm 0.0181$ |
| `no_inv` | Coord. decoder | $0.8511 \pm 0.0126$ | $0.0335 \pm 0.0014$ | $\mathbf{0.0358 \pm 0.0059}$ |
| | GA rotor | $\mathbf{0.9139 \pm 0.0259}$ | $\mathbf{0.0104 \pm 0.0009}$ | $0.0748 \pm 0.0072$ |
| `no_can_dice` | Coord. decoder | $0.8396 \pm 0.0098$ | $0.0292 \pm 0.0013$ | $\mathbf{0.0510 \pm 0.0054}$ |
| | GA rotor | $\mathbf{0.8904 \pm 0.0342}$ | $\mathbf{0.0100 \pm 0.0009}$ | $0.0900 \pm 0.0122$ |
| `no_view_dice` | Coord. decoder | $0.8222 \pm 0.0083$ | $0.0332 \pm 0.0026$ | $\mathbf{0.0586 \pm 0.0075}$ |
| | GA rotor | $\mathbf{0.8877 \pm 0.0355}$ | $\mathbf{0.0096 \pm 0.0008}$ | $0.0906 \pm 0.0148$ |
| `no_can_supervision` | Coord. decoder | $0.8412 \pm 0.0168$ | $0.0252 \pm 0.0009$ | $\mathbf{0.0498 \pm 0.0047}$ |
| | GA rotor | $\mathbf{0.8628 \pm 0.0220}$ | $\mathbf{0.0105 \pm 0.0008}$ | $0.0897 \pm 0.0070$ |
| `bce_only` | Coord. decoder | $0.8439 \pm 0.0158$ | $0.0262 \pm 0.0008$ | $\mathbf{0.0718 \pm 0.0075}$ |
| | GA rotor | $\mathbf{0.8890 \pm 0.0382}$ | $\mathbf{0.0100 \pm 0.0008}$ | $0.1024 \pm 0.0200$ |

correspond to the following settings, `full` uses all terms in equation 5, `no_conf` sets $\lambda_{\mathrm{conf}} = 0$, `no_inv` sets $\lambda_{\mathrm{inv}} = 0$, `no_can_dice` sets $\lambda_{\mathrm{can\text{-}Dice}} = 0$, `no_view_dice` sets $\lambda_{\mathrm{view\text{-}Dice}} = 0$, `no_can_supervision` sets both $\lambda_{\mathrm{can\text{-}BCE}} = 0$ and $\lambda_{\mathrm{can\text{-}Dice}} = 0$ and `bce_only` keeps observed-view BCE, canonical BCE with unit weight and the small KL term while removing Dice, invariance and confidence penalties. The BCE and KL terms are the VAE reconstruction and latent regularisation components (Kingma & Welling, 2014), the Dice terms align training with binary occupancy overlap (Milletari et al., 2016), the invariance term encourages the two observed views to encode the same content and the confidence term is the Bernoulli variance proxy $p(1-p)$, which suppresses ambiguous occupancies near probability $1/2$. Removing the confidence penalty leaves the ordering essentially unchanged with GA rotor view Dice at $0.9033 \pm 0.0316$ and composition MSE at $0.0101 \pm 0.0009$. This matters because the confidence penalty directly affects prediction sharpness. Removing the invariance term also preserves the same qualitative ordering with GA rotor view Dice at $0.9139 \pm 0.0259$ and composition MSE at $0.0104 \pm 0.0009$, which suggests that the relative-pose consistency is not only a consequence of forcing the two encoder means to match. The `no_can_supervision` row is diagnostic for posed-view and composition behaviour, but it is not proposed as a replacement canonicalisation objective because it removes the canonical anchor. The entropy column is also informative, the coordinate decoder often has lower entropy than the rotor model, so the rotor's view-Dice advantage is not explained by the rotor predictions simply being sharper.

## 5.4 A controlled 3$D$ voxel experiment preserves the composition pattern

Table 7: Controlled 3$D$ voxel experiment over ten paired seeds and 24 training epochs. Best values are bold. View BCE is the sampled latent two-view validation BCE sum. Thresholded Dice, canonical metrics and composition MSE use deterministic mean path logits. The exact-action row uses the ground truth SO(3) matrix induced by the sampled unit quaternion/spatial rotor for trilinear inverse sampling (it is not a separate rotor versus matrix comparison).

| Model | Canonical BCE ↓ | Canonical Dice ↑ | View BCE ↓ | View Dice ↑ | Composition MSE ↓ |
|---|---|---|---|---|---|
| Coord. decoder | $\mathbf{0.1284 \pm 0.0043}$ | $\mathbf{0.6227 \pm 0.0085}$ | $\mathbf{0.2964 \pm 0.0129}$ | $0.6089 \pm 0.0081$ | $0.0482 \pm 0.0011$ |
| Learned warp | $0.1306 \pm 0.0052$ | $0.6191 \pm 0.0108$ | $0.3006 \pm 0.0103$ | $0.5796 \pm 0.0133$ | $0.0524 \pm 0.0020$ |
| Exact SO(3) action | $0.1295 \pm 0.0054$ | $0.6219 \pm 0.0086$ | $0.5157 \pm 0.0132$ | $\mathbf{0.6207 \pm 0.0084}$ | $\mathbf{0.0202 \pm 0.0008}$ |

The 3$D$ result in Table 7 is a rotation only known-pose test. The table contains one exact-action row because a unit quaternion/spatial rotor and its induced SO(3) matrix define the same coordinate warp at the sampler. This is not a hidden rotor versus matrix comparison. It is one exact SO(3) action implemented through the

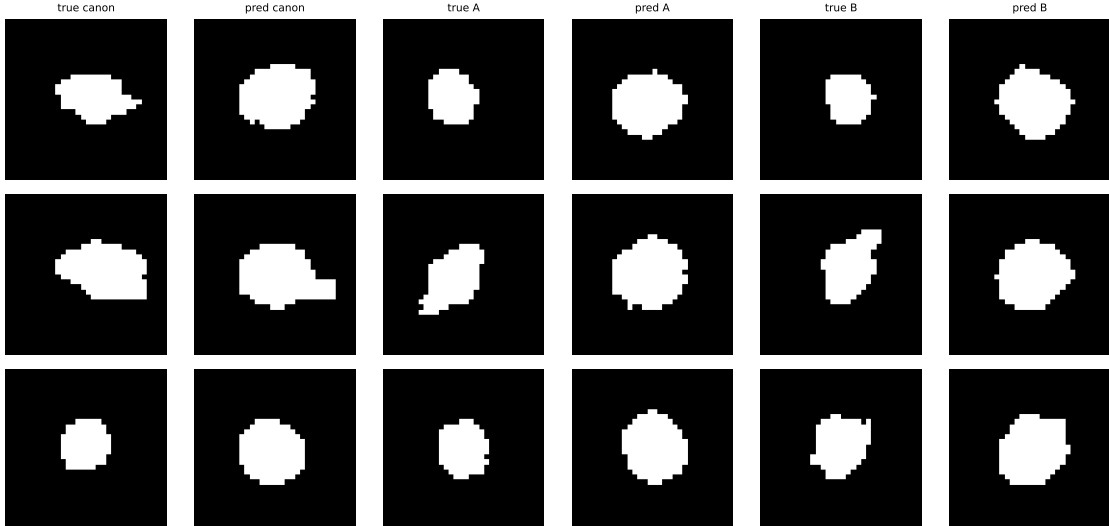

Figure 3: Qualitative slices from the $3D$ voxel experiment for a single validation example. Each row shows a central $2D$ slice along a different axis ($z$, $y$ and $x$) through the $32^3$ voxel grid, and each column corresponds to a different volume type. True canon and pred canon are the ground truth and predicted canonical occupancies. True A, pred A, true B and pred B are the two ground truth posed-views and their model predictions under the supplied multi-axis rotations. The figure illustrates geometrically consistent but coarse occupancy reconstruction rather than photorealistic $3D$ rendering.

matrix interface required by the sampler with the unit quaternion used to sample and parameterise the known rotation. The exact-action row reduces composition MSE from $0.0482 \pm 0.0011$ for the coordinate decoder to $0.0202 \pm 0.0008$ and it gives a modestly higher view Dice, $0.6207 \pm 0.0084$ versus $0.6089 \pm 0.0081$. However, the coordinate decoder obtains the best canonical BCE, canonical Dice and sampled latent View BCE diagnostic. The $3D$ study therefore strengthens the claim that exact-actions impose relative-pose consistency under multi-axis rotations, but it does not show broad $3D$ reconstruction superiority. Figure 3 shows the same boundary visually, because the posed outputs are geometrically consistent but remain coarse voxel reconstructions. Appendix B explains the rotor, quaternion and matrix relationship and why motors apply to future full rigid motion experiments with translations. Appendix C reports the independent `clifford` verification. Appendix H reports compact per-seed numerical summaries. Appendix I contains the main $2D$ metric summary, representative $2D$ reconstructions, the loss ablation view Dice summary and representative Rotated MNIST reconstructions.

## 6 Discussion and Limitations

The experiments support a calibrated conclusion. In controlled known-pose VAEs, applying the supplied pose through an exact-action improves thresholded posed-view fidelity and relative-pose consistency in the main $2D$ setting. The effect is not reproduced by a learned-warp control and it extends to angle generalisation, Rotated MNIST and $3D$ composition tests. At the same time, the coordinate decoder can obtain a lower sampled latent View BCE diagnostic and better canonical reconstruction after the longer training budget, and scaling shows that the view Dice gap can shrink or become comparable when data and capacity increase. These observations are consistent with the intended inductive bias interpretation. Exact-actions are most useful when the transformation law is known and should not be relearned from limited data, but they are not a guarantee of better values on every reconstruction metric.

Several boundaries remain. First, pose is supplied, so the paper does not solve pose inference. Second, the main $2D$ dataset has canonical supervision, so the paper does not solve unsupervised canonical frame discovery. Third, the datasets use binary occupancies without texture, lighting, shadows or background variation. In

real images, rotation can change shading and visibility and a coordinate-conditioned neural renderer may be useful precisely because it can learn angle dependent appearance effects that a fixed spatial sampler cannot express. Our results therefore apply most directly to settings where the known transformation is a geometric warp of occupancy or intensity fields and they should not be read as evidence against learned rendering when illumination and texture are part of the pose-conditioned appearance. Fourth, the $3D$ experiment uses voxel occupancy rather than differentiable photorealistic rendering. Fifth, the shared latent posterior is a controlled comparison device, whereas mixture of experts or product of experts objectives would provide a more principled multimodal variational formulation (Shi et al., 2019; Sutter et al., 2021).

Finally, planar GA is algebraically equivalent to an SO(2) matrix or a rotation only spatial transformer and the $3D$ rotation only action is represented at the sampler by the SO(3) matrix induced by a sampled unit quaternion/spatial rotor. This equivalence is used as a transparency control and is independently verified in Appendix C. A genuine motor experiment would require full rigid motions, such as coupled rotations and translations in SE(3), which is a different task from the centred rotation study reported here and is a natural extension of the rotor and registration literature (Doran & Lasenby, 2003; Lasenby et al., 2024; Matsantonis & Lasenby, 2022; 2023; Matsantonis, 2025).

## 7 Conclusion

This paper isolates a concrete design choice in known-pose generative modelling. A decoder can learn the effect of pose from coordinates, or the model can apply the known pose through an exact analytic action after decoding canonical content. The results show that the exact-action pathway gives stronger thresholded shape fidelity and much lower composition error in a controlled $2D$ paired-view VAE, that this effect is not explained by merely adding a learned warp layer and that the composition advantage persists in Rotated MNIST and in a minimal $3D$ voxel setting. The appropriate takeaway is therefore not that planar rotors are better than matrices or that rotation only voxels require motors. GA provides the unifying action notation and a route to future SE(3) motor experiments, while the present empirical claim is about fixed known-pose actions versus learned coordinate-conditioned rendering. Known transformations should be treated as transformations when the modelling goal is to separate canonical content from supplied pose.

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

## A    Planar Rotor Derivation and Raster Sign Convention

This appendix spells out the planar rotor calculation used by the analytic GA row. Let $e_1, e_2$ be orthonormal basis vectors of $\mathrm{Cl}(2,0)$, so $e_1^2 = e_2^2 = 1$ and $e_1e_2 = -e_2e_1$. Define the unit bivector $\mathbf{I} = e_1e_2$. Then:

$$\mathbf{I}^2 = (e_1e_2)(e_1e_2) \tag{7}$$
$$= e_1(e_2e_1)e_2 \tag{8}$$
$$= e_1(-e_1e_2)e_2 \tag{9}$$
$$= -e_1e_1e_2e_2 \tag{10}$$
$$= -1 \tag{11}$$

For an active Euclidean angle $\phi$, write $c = \cos(\phi/2)$ and $s = \sin(\phi/2)$. The rotor and reverse are:

$$r(\phi) = c - s\mathbf{I}, \qquad r(\phi)^\dagger = c + s\mathbf{I} \tag{12}$$

For a vector $u = u_1e_1 + u_2e_2$, first use $\mathbf{I}e_1 = -e_2$ and $\mathbf{I}e_2 = e_1$ to compute the left product:

$$r(\phi)u = (c - s\mathbf{I})(u_1e_1 + u_2e_2) \tag{13}$$
$$= cu_1e_1 + cu_2e_2 - su_1\mathbf{I}e_1 - su_2\mathbf{I}e_2 \tag{14}$$
$$= cu_1e_1 + cu_2e_2 - su_1(-e_2) - su_2(e_1) \tag{15}$$
$$= (cu_1 - su_2)e_1 + (cu_2 + su_1)e_2 \tag{16}$$

Now set $a = cu_1 - su_2$ and $b = cu_2 + su_1$. Since $e_1\mathbf{I} = e_2$ and $e_2\mathbf{I} = -e_1$, right multiplication by the reverse gives:

$$r(\phi)ur(\phi)^\dagger = (ae_1 + be_2)(c + s\mathbf{I}) \tag{17}$$
$$= ace_1 + bce_2 + ase_1\mathbf{I} + bse_2\mathbf{I} \tag{18}$$
$$= ace_1 + bce_2 + ase_2 - bse_1 \tag{19}$$
$$= (ac - bs)e_1 + (bc + as)e_2 \tag{20}$$

Substituting $a$ and $b$ gives:

$$ac - bs = c(cu_1 - su_2) - s(cu_2 + su_1) \tag{21}$$
$$= (c^2 - s^2)u_1 - 2csu_2, \tag{22}$$
$$bc + as = c(cu_2 + su_1) + s(cu_1 - su_2) \tag{23}$$
$$= 2csu_1 + (c^2 - s^2)u_2 \tag{24}$$

Finally, using $c^2 - s^2 = \cos\phi$ and $2cs = \sin\phi$, the sandwich action is:

$$r(\phi)ur(\phi)^\dagger = (\cos\phi\, u_1 - \sin\phi\, u_2)e_1 + (\sin\phi\, u_1 + \cos\phi\, u_2)e_2 \tag{25}$$

Therefore the induced matrix is the standard active rotation:

$$R(\phi) = \begin{bmatrix} \cos\phi & -\sin\phi \\ \sin\phi & \cos\phi \end{bmatrix} \tag{26}$$

The raster sign follows from the downward image $y$-axis. If an image grid coordinate is $q = (q_x, q_y)$ with $q_y$ increasing downward, the corresponding Euclidean coordinate is $u = (q_x, -q_y)$. Applying $R(\phi)$ in Euclidean coordinates and converting back to image coordinates gives:

$$u' = R(\phi)u \tag{27}$$
$$= (\cos\phi\, q_x + \sin\phi\, q_y,\ \sin\phi\, q_x - \cos\phi\, q_y), \tag{28}$$
$$q' = (u_1', -u_2') \tag{29}$$
$$= (\cos\phi\, q_x + \sin\phi\, q_y,\ -\sin\phi\, q_x + \cos\phi\, q_y) \tag{30}$$

Thus a positive Euclidean rotation by $\phi$ corresponds to an image-coordinate rotation with the opposite sign convention. In the controlled $2D$ silhouette experiments, a positive raster angle $\theta$ is therefore converted to the Euclidean rotor $r(-\theta)$ before applying the exact-action. The Rotated MNIST script uses a sampler-coordinate angle convention internally, i.e. target generation and exact-action rendering use the same PyTorch inverse sampling operator, so that robustness check is self-consistent under its own convention.

## B  $3D$ Rotations, Spatial Rotors and the Sampling Interface

The $3D$ extension uses rotations about the centre of the voxel grid. This is an SO(3) task rather than a full SE(3) rigid motion task. Spatial rotors in $\mathrm{Cl}(3,0)$, unit quaternions and rotation matrices are different coordinate descriptions of the same known rotation, whereas conformal or projective motors become necessary when the task includes translations or general rigid-body motions (Doran & Lasenby, 2003; Dorst et al., 2007; Lasenby et al., 2024; Matsantonis, 2025). The experiment therefore reports one exact SO(3) analytic action row rather than separate rotor and matrix rows.

Let $e_1, e_2, e_3$ be an orthonormal basis of $\mathrm{Cl}(3,0)$. The even subalgebra has basis $1, e_2e_3, e_3e_1, e_1e_2$. A unit spatial rotor can therefore be written as:

$$r = q_0 - q_1 e_2 e_3 - q_2 e_3 e_1 - q_3 e_1 e_2 \qquad q_0^2 + q_1^2 + q_2^2 + q_3^2 = 1 \tag{31}$$

The coefficient tuple $q = (q_0, q_1, q_2, q_3)$ is the usual unit-quaternion coordinate representation of this rotor, up to the chosen sign convention for the bivector basis. For a vector $u = u_1 e_1 + u_2 e_2 + u_3 e_3$, the analytic rotation is the sandwich action:

$$u' = r u r^\dagger \tag{32}$$

The voxel sampler does not consume multivectors. It consumes affine coordinate maps. In the implementation, the data generator samples unit-quaternion or spatial rotor coefficients and stores the induced matrix

$$R(q) = \begin{bmatrix} 1 - 2q_2^2 - 2q_3^2 & 2q_1 q_2 - 2q_3 q_0 & 2q_1 q_3 + 2q_2 q_0 \\ 2q_1 q_2 + 2q_3 q_0 & 1 - 2q_1^2 - 2q_3^2 & 2q_2 q_3 - 2q_1 q_0 \\ 2q_1 q_3 - 2q_2 q_0 & 2q_2 q_3 + 2q_1 q_0 & 1 - 2q_1^2 - 2q_2^2 \end{bmatrix} \tag{33}$$

The inverse sampling warp then evaluates the canonical voxel logit field $\ell_{\mathrm{can}}$ at $R(q)^{-1} u = R(q)^\top u$:

$$\ell_q(u) = \ell_{\mathrm{can}}\big(R(q)^\top u\big) \tag{34}$$

This conversion is a numerical interface, not a learned matrix baseline. The analytic $3D$ model utilizes the resulting ground truth matrix during training and evaluation because this is the format required by the sampler. The distinction tested in $3D$ is the same as in $2D$, exact known-pose action versus coordinate-conditioned or learned-warp rendering.

A conformal motor experiment would be a different controlled problem, as in such a setting the pose would include translations as well as rotations, and the action would be a motor sandwich in a conformal or projective algebra before being converted to the affine sampler. *We leave that* SE(3) *extension for future work because adding translations would change the data generation process, boundary effects and identifiability conditions.*

## C  Independent `clifford` Verification of $2D$ and $3D$ Rotor Calculations

The experiments implement the induced matrices because PyTorch grid samplers consume affine coordinate maps. To verify that these matrices match the Geometric Algebra formulas, we used the Python `clifford` package (Hadfield et al., 2021) to construct $\mathrm{Cl}(2,0)$ and $\mathrm{Cl}(3,0)$ multivectors, then compare the rotor sandwich action with the matrix action used in the code. It checks the $2D$ planar rotor $r(\theta) = \cos(\theta/2) - \sin(\theta/2)\mathbf{I}$, $2D$ rotor composition, the $3D$ spatial rotor $r = q_0 - q_1 e_2 e_3 - q_2 e_3 e_1 - q_3 e_1 e_2$, and $3D$ composition under the Hamilton product convention used to generate the voxel rotations.

| Check | $2D$ maximum absolute error | $3D$ maximum absolute error |
|---|---|---|
| Rotor unit norm | $2.22 \times 10^{-16}$ | $4.44 \times 10^{-16}$ |
| Sandwich action versus induced matrix | $8.88 \times 10^{-16}$ | $1.33 \times 10^{-15}$ |
| Rotor-product composition action | $1.22 \times 10^{-15}$ | $2.22 \times 10^{-15}$ |
| Matrix composition consistency | $5.13 \times 10^{-16}$ | $1.11 \times 10^{-15}$ |

Table A1: Independent `clifford` verification over 1000 random tests per block with seed 12345. The reported discrepancies are at floating point roundoff scale. The $3D$ check verifies the spatial-rotor/unit-quaternion convention used by the voxel sampler (it is not a separate learned $3D$ GA model).

These checks support the algebraic implementation claims only. They do not turn the $2D$ or $3D$ experiments into rotor versus matrix comparisons, because the corresponding rotor and matrix actions are isomorphic coordinate descriptions of the same known rotations. Table A1 summarises the independent numerical checks used to verify that the rotor sandwich actions and the induced matrices agree to floating point precision.

## D  Canonical Supervision and Asymmetric Shapes

The experiments evaluate both posed views and the canonical prediction, so all models receive the same auxiliary canonical target. This is especially important for the coordinate decoder, i.e. its canonical prediction is read by setting the pose code to $p(0)$, while the posed-view losses only train outputs at the observed angles $p(\theta_a)$ and $p(\theta_b)$. Without the canonical term, the decoder could reconstruct the two observed views while leaving the zero angle output poorly constrained.

The fixed and procedural polygon datasets use asymmetric shapes to avoid rotational stabilisers. If a canonical object is invariant under a non-trivial rotation, then multiple angles can produce the same raster target. In that case, a direct canonical orientation metric would mix reconstruction error with orientation non-identifiability. The asymmetric templates and random asymmetric polygons are therefore a measurement choice, not an assumption about natural objects.

## E  Metrics, Loss and Statistical Reporting

For a logit field $\ell$ and binary target $y$, the mean Bernoulli binary cross-entropy is:

$$\mathrm{BCE}(\ell, y) = -\frac{1}{|\Omega|} \sum_{u \in \Omega} [y(u) \log \mathrm{sigm}(\ell(u)) + (1 - y(u)) \log(1 - \mathrm{sigm}(\ell(u)))] \tag{35}$$

where $\Omega$ is the pixel or voxel grid and $\mathrm{sigm}(t) = (1 + \exp(-t))^{-1}$. The soft Dice coefficient used in the training loss is:

$$\mathrm{Dice}_{\mathrm{soft}}(\ell, y) = \frac{2 \sum_{u \in \Omega} \mathrm{sigm}(\ell(u)) y(u) + \eta}{\sum_{u \in \Omega} \mathrm{sigm}(\ell(u)) + \sum_{u \in \Omega} y(u) + \eta} \tag{36}$$

with small constant $\eta > 0$. The thresholded Dice coefficient reported in the tables replaces $\mathrm{sigm}(\ell(u))$ by $\mathbf{1}\{\mathrm{sigm}(\ell(u)) > 1/2\}$. If $\hat{\ell}_a$ and $\hat{\ell}_b$ are the two predicted view logits for the same example, and $G_{b \leftarrow a}$ is the known relative pose action from view $a$ to view $b$, the composition metric is:

$$\mathrm{MSE}_{\mathrm{comp}} = \frac{1}{|\Omega|} \sum_{u \in \Omega} \left( \mathrm{sigm}\left((G_{b \leftarrow a} \hat{\ell}_a)(u)\right) - \mathrm{sigm}(\hat{\ell}_b(u)) \right)^2 \tag{37}$$

For $2D$ rotations, $G_{b \leftarrow a}$ is induced by the relative angle $\theta_b - \theta_a$ with the raster sign correction above. For $3D$ rotations, $G_{b \leftarrow a}$ is induced by $R_b R_a^\top$.

The full training loss is shared by the compared models. Let $\ell_a^z, \ell_b^z$ be the sampled latent view logits used in the VAE reconstruction objective, let $\ell_{\mathrm{can}}^\mu, \ell_a^\mu, \ell_b^\mu$ be the corresponding deterministic mean path logits and let

$q_a(z), q_b(z)$ be the two Gaussian content posteriors with means $\mu_a, \mu_b$. The implemented loss is:

$$\mathcal{L} = \text{BCE}(\ell_a^z, x_a) + \text{BCE}(\ell_b^z, x_b) \tag{38}$$

$$+ \lambda_{\text{view-Dice}} \left[ 2 - \text{Dice}_{\text{soft}}(\ell_a^z, x_a) - \text{Dice}_{\text{soft}}(\ell_b^z, x_b) \right] \tag{39}$$

$$+ \lambda_{\text{can-BCE}} \text{BCE}(\ell_{\text{can}}^\mu, c) + \lambda_{\text{can-Dice}} \left[ 1 - \text{Dice}_{\text{soft}}(\ell_{\text{can}}^\mu, c) \right] \tag{40}$$

$$+ \lambda_{\text{inv}} \frac{1}{d_z} \|\mu_a - \mu_b\|_2^2 + \beta_z \left[ D_{\text{KL}}(q_a(z) \| \mathcal{N}(0, I)) + D_{\text{KL}}(q_b(z) \| \mathcal{N}(0, I)) \right] \tag{41}$$

$$+ \lambda_{\text{conf}} \mathcal{L}_{\text{conf}} \tag{42}$$

Here $d_z$ is the latent dimension, the invariance term is therefore the elementwise mean squared difference between the two content means. The confidence term is implemented as the sum of three fieldwise mean penalties:

$$
\begin{aligned}
\mathcal{L}_{\text{conf}} = &\frac{1}{|\Omega|} \sum_{u \in \Omega} \text{sigm}(\ell_{\text{can}}^\mu(u))(1 - \text{sigm}(\ell_{\text{can}}^\mu(u))) \\
&+ \frac{1}{|\Omega|} \sum_{u \in \Omega} \text{sigm}(\ell_a^\mu(u))(1 - \text{sigm}(\ell_a^\mu(u))) \\
&+ \frac{1}{|\Omega|} \sum_{u \in \Omega} \text{sigm}(\ell_b^\mu(u))(1 - \text{sigm}(\ell_b^\mu(u)))
\end{aligned}
\tag{43}
$$

The reported View BCE is $\text{BCE}(\ell_a^z, x_a) + \text{BCE}(\ell_b^z, x_b)$ on the validation pass, so it is a sampled latent two-view reconstruction diagnostic. The reported canonical BCE/Dice, thresholded view Dice and composition MSE use deterministic mean path logits. We therefore place the main geometric emphasis on view Dice and composition consistency rather than on View BCE alone. The main loss weights are $\lambda_{\text{can-BCE}} = 5$, $\lambda_{\text{can-Dice}} = 2$, $\lambda_{\text{view-Dice}} = 1$, $\lambda_{\text{inv}} = 5$, $\beta_z = 10^{-4}$ and $\lambda_{\text{conf}} = 0.02$.

The loss terms have the following roles. BCE is the Bernoulli negative log-likelihood used for binary VAE reconstruction and the Kullback-Leibler (KL) term is the standard Gaussian latent regulariser in VAEs (Kingma & Welling, 2014). Dice is included because the target objects are binary occupancies and Dice overlap is a direct shape-agreement metric (Milletari et al., 2016). Canonical BCE/Dice are supervised auxiliary losses used for both decoder types because the canonical output is evaluated directly and because posed-view reconstruction alone does not anchor the absolute canonical frame. The invariance loss encourages the two observed views of the same object to map to the same content mean through an elementwise MSE penalty. The confidence term is not a separate geometric assumption; it is the Bernoulli variance proxy $p(1 - p)$, minimised at $p = 0$ or $p = 1$ and maximised at $p = 1/2$, so it penalizes uncertain occupancy probabilities.

## F Architecture, Data and Optimisation Details

The $2D$ encoder has four stride-2 convolutional layers with widths $C, 2C, 4C, 8C$, SiLU (sigmoid linear unit) activations, a flattening layer, a 256-unit linear layer and two linear heads for $\mu$ and $\log \sigma^2$. The canonical decoder mirrors this structure with a 256-unit linear layer, reshaping to an $8C$ feature map, and four transposed convolution layers. The coordinate decoder receives $d + 2$ inputs, because the angle is encoded as $[\cos\theta, \sin\theta]$. The $3D$ coordinate decoder receives $d + 6$ inputs using the continuous $6D$ rotation representation of Zhou et al. (2019). The $3D$ exact-action decoder receives no learned pose coordinates, i.e. it decodes one canonical voxel field and applies the ground truth SO(3) matrix induced by the sampled unit quaternion/spatial rotor as described in Appendix B. The learned-warp baselines use the same canonical decoder and add a small pose network to predict constrained rotation parameters before sampling.

The fixed $2D$ canonical silhouettes are generated from six asymmetric polygon templates, rasterised at eight times the final resolution, downsampled bilinearly to $32 \times 32$ and thresholded at 0.5. In the controlled $2D$ silhouette experiments, the posed target views are generated by Python Imaging Library (PIL) bilinear rotation and thresholding, whereas the exact-action decoders apply their analytic pose action through PyTorch inverse grid sampling. This fixed sampler difference is not tuned per model, i.e. it should be interpreted as a small implementation level interpolation mismatch rather than as a learned component of any method.

The procedural shape experiment replaces these templates by random star-like asymmetric polygons. The Rotated MNIST experiment draws from the standard MNIST train and test partitions, converts upright digits to binary occupancy by thresholding normalised intensities at 0.15, pads them from $28 \times 28$ to $32 \times 32$ and generates two independent rotated views with angles sampled from $[-\pi, \pi)$. The Rotated MNIST targets are produced by the same PyTorch bilinear inverse sampling operator used by the exact-action models and then thresholded at 0.25, so this robustness check is internally consistent under its sampler-coordinate angle convention. These thresholds are fixed for all models and seeds; they are not tuned per method. The $3D$ canonical volumes are asymmetric unions of simple cuboids and ellipsoids on a $32^3$ grid. These choices are not meant to mimic all natural image complexity, but they keep the pose action, canonical target and relative composition measurable while gradually increasing data variation.

The main $2D$ setting uses $N_{\text{train}} = 256$, $N_{\text{val}} = 128$, batch size 32, 32 epochs, Adam with learning rate $2 \times 10^{-3}$, $d = 16$, $C = 8$ and seeds $0, \ldots, 9$. Scaling runs vary only $N$, $d$, and $C$ and also use 32 epochs. Angle generalisation and procedural shape runs use $N_{\text{train}} = 512$ and $N_{\text{val}} = 256$ so that each setting has enough examples across the allowed shapes or angles, again with 32 epochs. Rotated MNIST uses $N_{\text{train}} = 12,000$, $N_{\text{val}} = 2,000$, batch size 256, 32 epochs, Adam with learning rate $2 \times 10^{-3}$, $d = 32$, $C = 16$ and seeds $0, \ldots, 9$. The $3D$ setting uses $N_{\text{train}} = 512$, $N_{\text{val}} = 128$, batch size 8, 24 epochs, Adam with learning rate $10^{-3}$, $d = 32$, $C = 8$ and seeds $0, \ldots, 9$. No setting uses per-model hyperparameter tuning. This fixed protocol choice is part of the experimental design because all models are meant to differ in pose pathway rather than in tuning budget.

## G    Rationale for Model and Dataset Choices

Table A2 summarises why each pathway and dataset is included in the controlled comparison.

Table A2: Rationale for the model and dataset choices. Each condition isolates one possible explanation of the observed effect rather than attempting to maximise benchmark performance.

| Choice | Why it is included | What it controls or tests |
|---|---|---|
| Coordinate decoder | Standard neural conditioning by $[\cos\theta, \sin\theta]$ gives a direct learned-rendering baseline. | Tests whether the decoder can learn the pose action from coordinates. |
| Learned warp | Uses a differentiable sampler but predicts the warp from pose coordinates. | Separates the value of a warp layer from the value of the exact known action. |
| Exact matrix/spatial transformer network (STN) | Applies the supplied rotation with the same inverse sampling convention as a rotation-only spatial transformer (Jaderberg et al., 2015). | Checks that the $2D$ result is about analytic action and not a GA-specific advantage over matrices. |
| GA rotor and spatial rotor | Builds the $2D$ action from planar rotor coefficients. In $3D$ rotations are sampled as unit quaternions which are coordinate representations of spatial rotors and the analytic model uses the induced SO(3) matrices for sampling. | Provides the proposed exact action pathway without claiming matrix inferiority in rotation-only raster warping. |
| Fixed silhouettes | Six asymmetric templates give exact canonical targets and identifiable orientation. | Creates a low data controlled setting where non-pose variation is limited. |
| Procedural polygons and Rotated MNIST | Random polygons remove the fixed template restriction, and Rotated MNIST uses real digit shapes with known synthetic rotations (LeCun et al., 1998). | Tests whether the observed pattern is specific to six hand designed silhouettes. |
| 3D voxels | Known multi-axis rotations act on binary occupancies without a learned photorealistic renderer. | Tests the same pathway question in a non-commutative rotation group while avoiding an unsupported motor or full SE(3) claim. |

## H    Per-Seed Values and Paired Tests

Tables A3-A5 show the seed level pattern behind the aggregate means. In the main $2D$ task, the GA rotor has higher view Dice and lower composition MSE than the coordinate decoder in all ten displayed seeds. On Rotated MNIST, the GA rotor has lower composition MSE in all ten seeds and higher view Dice in seven of ten seeds which is why the MNIST result is interpreted mainly as support for the composition consistency pattern rather than as a universal reconstruction advantage. In the $3D$ voxel task, the exact SO(3) action has lower composition MSE in all ten seeds and higher view Dice in nine of ten seeds with the remaining seed effectively tied.

Table A3: Key per-seed 2$D$ values. Dice is view Dice and Comp. is composition MSE. Bold entries indicate the best displayed value within each seed and metric.

| Seed | Coord. Dice | GA Dice | Coord. Comp. | GA Comp. |
|------|-------------|---------|--------------|----------|
| 0 | 0.8309 | **0.8875** | 0.0293 | **0.0104** |
| 1 | 0.8139 | **0.8689** | 0.0355 | **0.0100** |
| 2 | 0.8485 | **0.9136** | 0.0328 | **0.0103** |
| 3 | 0.8292 | **0.9221** | 0.0299 | **0.0109** |
| 4 | 0.8445 | **0.8969** | 0.0316 | **0.0109** |
| 5 | 0.8548 | **0.9289** | 0.0351 | **0.0115** |
| 6 | 0.8301 | **0.8891** | 0.0370 | **0.0112** |
| 7 | 0.8701 | **0.8931** | 0.0290 | **0.0097** |
| 8 | 0.8533 | **0.9123** | 0.0353 | **0.0095** |
| 9 | 0.8302 | **0.9186** | 0.0312 | **0.0100** |

Table A4: Key per-seed Rotated MNIST values. Dice is view Dice and Comp. is composition MSE. Bold entries indicate the best displayed value within each seed and metric.

| Seed | Coord. Dice | GA Dice | Coord. Comp. | GA Comp. |
|------|-------------|---------|--------------|----------|
| 0 | 0.6827 | **0.8023** | 0.0265 | **0.0107** |
| 1 | **0.7995** | 0.7954 | 0.0309 | **0.0109** |
| 2 | 0.7799 | **0.7865** | 0.0302 | **0.0106** |
| 3 | 0.7964 | **0.8098** | 0.0309 | **0.0105** |
| 4 | 0.7933 | **0.7937** | 0.0305 | **0.0108** |
| 5 | 0.7020 | **0.7848** | 0.0273 | **0.0108** |
| 6 | 0.7824 | **0.7953** | 0.0315 | **0.0109** |
| 7 | **0.7943** | 0.7913 | 0.0312 | **0.0109** |
| 8 | 0.6761 | **0.7946** | 0.0270 | **0.0108** |
| 9 | **0.7953** | 0.7863 | 0.0311 | **0.0106** |

Table A5: Key per-seed 3$D$ values for the coordinate decoder and the single exact SO(3) action row. Dice is view Dice and Comp. is composition MSE. Bold entries indicate the best displayed value within each seed and metric.

| Seed | Coord. Dice | Exact Dice | Coord. Comp. | Exact Comp. |
|------|-------------|------------|--------------|-------------|
| 0 | 0.6017 | **0.6138** | 0.0473 | **0.0200** |
| 1 | **0.6120** | 0.6116 | 0.0500 | **0.0195** |
| 2 | 0.6243 | **0.6356** | 0.0471 | **0.0204** |
| 3 | 0.6143 | **0.6273** | 0.0485 | **0.0204** |
| 4 | 0.5949 | **0.6096** | 0.0469 | **0.0197** |
| 5 | 0.6069 | **0.6254** | 0.0475 | **0.0193** |
| 6 | 0.6089 | **0.6182** | 0.0479 | **0.0196** |
| 7 | 0.6138 | **0.6291** | 0.0495 | **0.0219** |
| 8 | 0.6028 | **0.6171** | 0.0493 | **0.0204** |
| 9 | 0.6096 | **0.6198** | 0.0485 | **0.0212** |

Table A6 reports paired tests for the main 2$D$ comparison. The sign is oriented so that positive differences favour the comparison model over the coordinate decoder after accounting for whether the metric is higher is better or lower is better. Canonical metrics favour the coordinate decoder and are not part of the positive analytic action claim, the robust positive effects are view Dice and composition MSE.

Table A6: Paired statistical tests for the main $2D$ comparison over ten seeds.

| Metric | Comparison | Mean diff | 95% CI | $p_t$ |
|---|---|---|---|---|
| Canonical BCE | Learned warp vs Coord. decoder | -0.0198 | [-0.0341, -0.0055] | 0.012 |
| Canonical BCE | Exact SO(2)/STN vs Coord. decoder | -0.0138 | [-0.0223, -0.0053] | 0.00511 |
| Canonical BCE | GA rotor vs Coord. decoder | -0.0125 | [-0.0229, -0.0020] | 0.0243 |
| Canonical Dice | Learned warp vs Coord. decoder | -0.0426 | [-0.0772, -0.0080] | 0.0212 |
| Canonical Dice | Exact SO(2)/STN vs Coord. decoder | -0.0244 | [-0.0421, -0.0068] | 0.012 |
| Canonical Dice | GA rotor vs Coord. decoder | -0.0210 | [-0.0418, -0.0003] | 0.0476 |
| View BCE | Learned warp vs Coord. decoder | -0.0690 | [-0.0885, -0.0495] | 2.17e-05 |
| View BCE | Exact SO(2)/STN vs Coord. decoder | -0.0321 | [-0.0484, -0.0159] | 0.00155 |
| View BCE | GA rotor vs Coord. decoder | -0.0295 | [-0.0471, -0.0119] | 0.00429 |
| View Dice | Learned warp vs Coord. decoder | -0.0595 | [-0.0813, -0.0377] | 0.000164 |
| View Dice | Exact SO(2)/STN vs Coord. decoder | 0.0598 | [0.0493, 0.0702] | 4.14e-07 |
| View Dice | GA rotor vs Coord. decoder | 0.0625 | [0.0484, 0.0767] | 3.5e-06 |
| Composition MSE | Learned warp vs Coord. decoder | -0.0109 | [-0.0181, -0.0037] | 0.00746 |
| Composition MSE | Exact SO(2)/STN vs Coord. decoder | 0.0224 | [0.0204, 0.0244] | 9.91e-10 |
| Composition MSE | GA rotor vs Coord. decoder | 0.0222 | [0.0202, 0.0242] | 1.28e-09 |

# I   Additional Visual Summaries

The figures in this appendix are included for interpretability rather than as additional quantitative claims. Figure A1 visualises the same ten-seed means and standard deviations as Table 2. Figure A2 visualises only the view-Dice part of Table 6, the table should be used for the corresponding composition and entropy values. Figure A3 shows representative thresholded $2D$ reconstructions. Figure A4 gives representative MNIST reconstructions to illustrate the greater shape variability of the digit task.

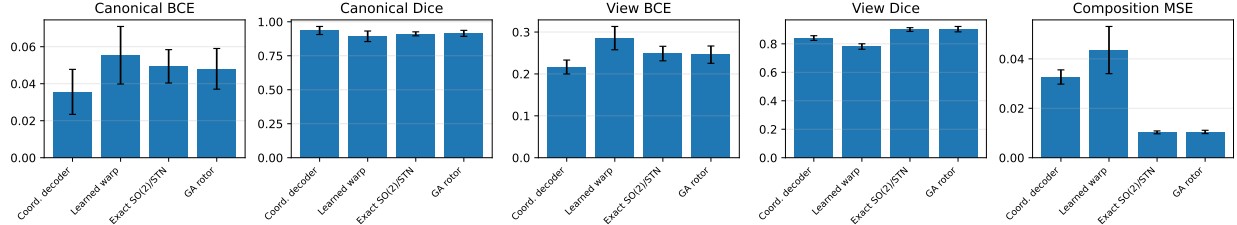

Figure A1: Metric summary for the ten-seed $2D$ comparison. This figure visualises the same means and standard deviations as Table 2.

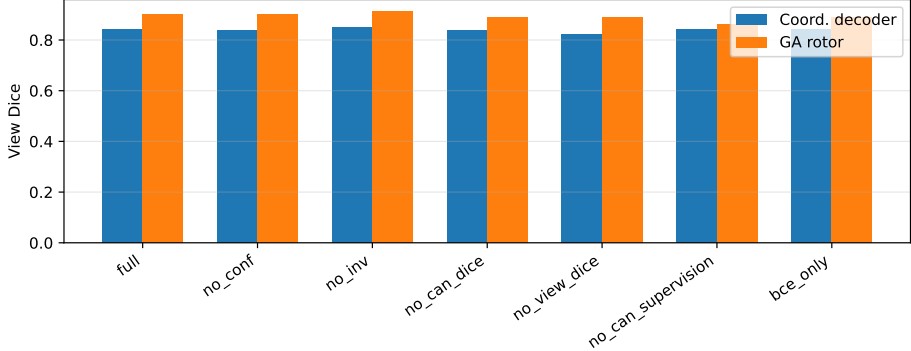

Figure A2: Loss ablation view Dice summary. Each bar corresponds to retraining the same decoder type under a different loss weight configuration, this is a loss ablation rather than a model architecture ablation. The main text table reports the corresponding composition MSE and entropy values.

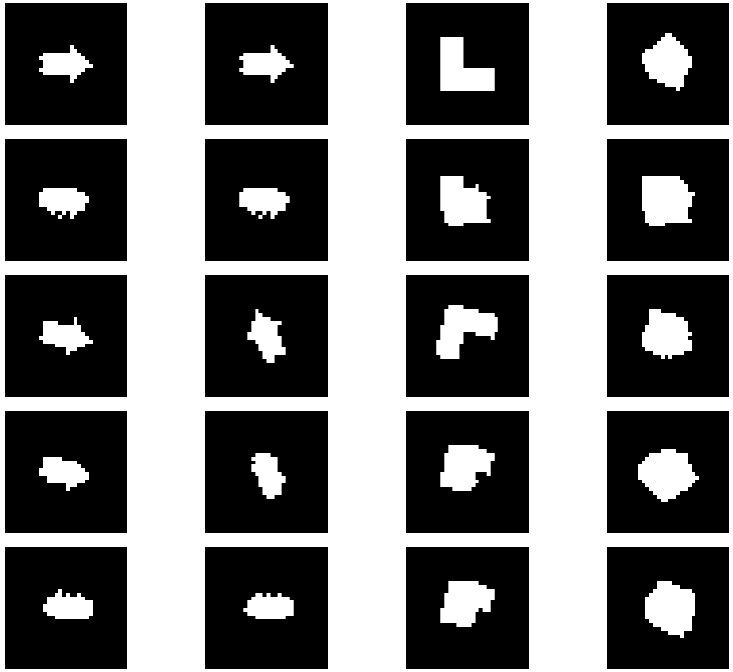

Figure A3: Representative $2D$ thresholded reconstructions. The quantitative claims rely on the ten-seed tables rather than this qualitative panel.

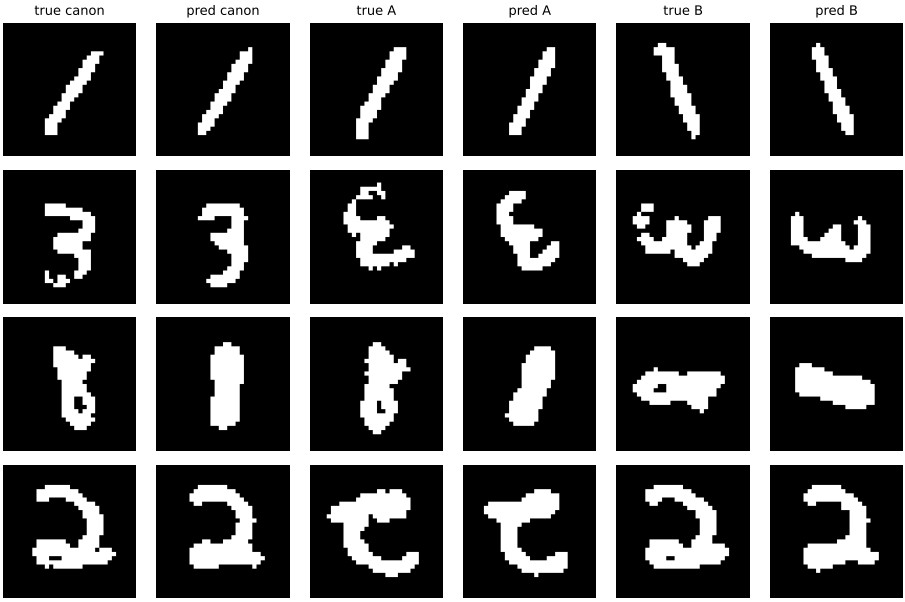

Figure A4: Representative Rotated MNIST reconstructions. The panel shows that digit variation makes the task less clean than the polygon experiment, which is why the MNIST result is interpreted mainly as a composition consistency check rather than as a universal reconstruction advantage.

