# OpenReview forum: "Exact Pose Actions versus Coordinate-Conditioned Rendering in Known-Pose Variational Autoencoders"
_TMLR — Under review for TMLR_

### Review · Reviewer_Fm6F · 2026-06-16

**Summary Of Contributions:**

**Summary**: The paper investigates the problem of building a special kind of pose-aware VAE, one where the encoder takes in rotated views to output representations of a "canonical" view, and the decoder takes this representation, and an angle input to output images rotated with that amount. In this setting, the paper investigates: is it better to apply pose through an exact geometric transformation or to concatenate pose coordinates to the decoder and let it learn the rendering rule? The main claim is that having the exact geometric transforms placed in the decoder help learning outcomes on this task.

**Strengths**

- The proposed method obtains a clearly improved composition-consistency MSE relative to the coordinate-conditioned learned-rendering baseline (0.0092 vs. 0.0319), which somewhat substantiates the paper's claim.

**Weaknesses**

- The paper does not motivate the problem well. The introduction opens by describing challenges (handling pose, composition rules of different pose representations) before establishing what problem is being solved or why it matters. It never makes clear whether the considered task is intended as a toy probe of geometric reasoning in VAEs or whether it arises in any real application. Since the setting assumes ground-truth pose is supplied and only canonical content must be learned, the practical relevance is unclear. Relatedly, the task is restricted to 2D rotation of canonical silhouettes, and the authors themselves acknowledge that the results do not extend to 3D or general multi-axis rotations.

- The experiments are extremely small in scale. The training set contains only 256 paired examples, which is surprising given that the data is fully synthetic and could be generated in arbitrary quantity; this leaves open whether the observed differences would persist with more data. The models are correspondingly tiny (four convolutional layers, latent dimensions of 16–64, base widths of 8–32), and the main quantitative claim rests on a deliberately compressed-capacity regime. While the authors argue this regime is chosen to expose inductive-bias differences, it raises the concern that the gap may shrink as capacity and data grow, which is precisely the regime a practitioner would care about.

- Finally, the method is evaluated in isolation. The paper introduces its own baseline and its own synthetic dataset, and does not refer to methods, baselines, or datasets from prior work, despite discussing spatial transformers, transformational autoencoders, equivariant VAEs in related work. The headline result is that the proposed model beats a naive baseline the paper itself constructs, so the broader significance—how this pathway compares to existing approaches the authors cite—remains unclear.

**Audience:**

No

**Audience Explanation:**

No. The paper does not motivate its task, and does not situate its work within the broader literature at all. Thus it is unlikely to be of interest to the general audience.

**Claims And Evidence:**

No

**Claims Explanation:**

The paper does not provide a single concrete falsifiable claim for the review to verify. The operating claim seems to be: "in the controlled known-pose 2D silhouette VAE, replacing coordinate-conditioned learned rendering with an explicit analytic warp improves thresholded shape fidelity and relative-pose composition consistency." However, the paper also hedges the claim so much so that the underlying claim is unclear. For example it states that it "does not claim planar rotors are superior to equivalent matrix or spatial-transformer warps" (which it concedes are mathematically identical in 2D), and that the composition metric "partly validates a property built into the pathway" rather than being an independent benchmark. Thus it is not clear what is precisely being claimed in this paper.

Even within the confines of the experiments, given the restricted toy nature of the experiments described in the summary; the experiments cannot be said to provide convincing or clear evidence for the underlying claim.

**Requested Changes:**

The paper requires a major revision / rewrite / reconceptualization in my review to be able to secure an acceptance. The main aspects to tackle are:
- stating a clear, falsifiable claim that can be verified
- rewriting the paper to state a clear motivation: why is the problem being studied, and what is its significance?
- larger scale experiments with larger dataset and model sizes to provide clear and convincing evidence of the claims
- experiments on and comparison with existing methods / architectures / datasets

---

> ### Author Response · Authors · 2026-07-06
> **Response to Reviewer Fm6F**
>
> We thank the reviewer for the constructive assessment. The review made clear that the original version did not sufficiently motivate the problem or clearly state the claim. We are particularly grateful for the reviewer’s suggestions to add larger scale experiments, larger datasets/model capacities and stronger positioning relative to existing methods, architectural choices and datasets. We addressed these points by rewriting the motivation and claim, adding controlled architectural comparisons including a learned-warp control, adding finite data/capacity scaling, adding angle-generalisation and procedural shape checks, adding a Rotated MNIST known-pose robustness check and adding a controlled 3D voxel experiment. These suggestions gave us the opportunity to substantially improve the manuscript.
>
> 1. [Requested change]: State a clear, falsifiable claim.
>
> We revised the Abstract and Introduction. The Introduction now contains a boxed research question asking whether, in a matched known-pose paired-view VAE, routing supplied pose through an exact analytic action improves posed-view fidelity and relative-pose consistency compared with coordinate-conditioned decoding. We also clarify that the supported claim is an exact-action/sample efficiency claim, not universal reconstruction superiority. The Results identify the strongest supported metrics: thresholded view Dice and relative-pose composition MSE, while also reporting that the coordinate decoder obtains better sampled-latent View BCE and canonical BCE/Dice.
>
> 2. [Requested change]: Improve motivation and significance.
>
> We rewrote the Introduction around known-pose generative modelling. The revised text explains that pose metadata arise in synthetic rendering, robotic sensing, calibrated multi-view data and scientific acquisition protocols such as microscope stage rotation and tomography. In these cases the question is not always how to infer pose, but how a model should use pose that is already supplied, i.e. by learning the transformation rule from coordinates or by decoding canonical content and applying the known transformation exactly.
>
> 3. [Requested change]: Address the restricted 2D setting and lack of 3D evidence.
>
> The original version only sketched the possible 3D and motor extension. We added a controlled 3D voxel experiment introduced in Experimental Protocol and reported in the Results subsection. The exact-action row reduces composition MSE from 0.0482+-0.0011 to 0.0202+-0.0008 with a modest view-Dice improvement. We explicitly state there and in Appendix 3D Rotations, Spatial Rotors and the Sampling Interface that this is a rotation-only centred voxel experiment, not a full SE(3) motor experiment with translations.
>
> 4. [Requested change]: Add larger-scale experiments.
>
> We added the Results subsection Scaling, angle generalisation and procedural shapes test the scope of the effect, and the corresponding data/capacity scaling table. The grid uses $N \in \{256, 1024, 4096\}$ and $(d,C) \in \{(16,8), (32,16), (64,32)\}$ with independent splits and valid paired-seed reporting. We interpret this cautiously as a finite fixed protocol scaling test, i.e. the view-Dice gap can shrink with more data/capacity, while composition MSE remains lower for the exact-action row in the displayed valid cells. Appendix Architecture, Data and Optimisation Details gives the training budgets.
>
> 5. [Requested change]: Compare with methods, architectures and datasets.
>
> We addressed this within the controlled scope of the paper by adding architectural controls and dataset robustness checks, rather than a SOTA leaderboard that would confound the known-pose pathway question. Related Work now situates the study relative to VAEs, disentanglement, transforming autoencoders, spatial transformer networks, VITAE, group equivariant models, geometry-aware generative models, Clifford/GA networks, NeRF-VAE, EG3D and multimodal VAEs. Controlled Problem and Model Family defines the coordinate decoder, learned-warp control, exact SO(2)/rotation-only STN control and GA rotor pathway, while the 3D experiment reports one exact SO(3) action row. Appendix Rationale for Model and Dataset Choices explains the role of each comparison. We also added procedural asymmetric polygons and Rotated MNIST as robustness checks beyond the six fixed silhouette templates.
>
> 6. [Requested change]: Clarify whether this is a toy probe or practically relevant.
>
> We now state that the silhouette task is a controlled diagnostic, not a natural image benchmark. The Introduction explains that it isolates the pose pathway before adding texture, illumination, object detection, camera estimation or photorealistic rendering. Discussion and Limitations state that the conclusions apply most directly when the known transformation is a geometric warp of occupancy or intensity fields and should not be read as evidence against learned rendering when pose changes illumination, visibility or texture.

---

### Review · Reviewer_bMWS · 2026-06-22

**Summary Of Contributions:**

The paper's key contributions are:

 - Isolates one design choice in known-pose generative modeling: apply pose via a fixed analytic warp vs. concatenate pose coordinates to a learned decoder.

- Instantiates it in a deliberately minimal paired-view VAE for 2D binary silhouettes, with encoder, latent size, and decoder width matched across both models.

- Reports a three-seed compressed-capacity ablation: the analytic pathway improves thresholded view Dice and reduces relative-pose composition error to roughly one-third of the baseline.

- Provides full rotor algebra, raster sign-convention derivation, identifiability/asymmetry proofs.


Strengths
- Confounder isolation (capacity matched) and honest interpretation of the BCE-vs-Dice trade-off.

Weaknesses
- Narrow scope: one synthetic dataset (six hand-designed polygons), 32×32 binaries, three seeds, 12 epochs.

- The Geometric Algebra formalism is redundant in 2D and only earns its keep in a 3D extension that is sketched but never tested.

**Additional Comments:**

The paper is well-written and honest in its scoping. Its disclaimers and clean controlled-isolation methodology make it easy to read and are a genuine strength. The contribution is narrow, a single architectural comparison in a deliberately minimal 2D setting, but it is correct, clearly presented, and the authors are upfront about its limits.

**Audience:**

Yes

**Audience Explanation:**

- Researchers working on equivariant and geometry-aware generative models, spatial transformers, disentanglement, and geometric-algebra neural networks.

-The question the paper isolates, whether to apply a known transformation analytically or to let a decoder learn it from pose coordinates, is a clean instance of a recurring design choice, so the finding is relevant.

**Broader Impact Concerns:**

There are no significant ethical concerns that require a separate Broader Impact Statement.

**Claims And Evidence:**

Yes

**Claims Explanation:**

- The claims are scoped narrowly: the authors do not claim geometric algebra is a better tool than ordinary methods, and explicitly note that in 2D, a rotor, an SO(2) matrix, and a rotation-only spatial transformer compute the same rotation.

-  They make a narrower architectural claim: applying a known pose with a fixed, non-learned transformation beats handing the angle to the decoder and letting it learn the rendering from data. This is about fixed vs. learned action, not which formalism expresses it.

- The mathematical claims are derived in full and independently verified against the ```clifford``` library to floating-point precision.

- The scoped empirical claim is supported: the rotor pathway clearly improves thresholded view Dice and reduces relative-pose composition error to roughly a third of the baseline's.

- The one result that looks contrary, the baseline winning on probabilistic view BCE, is reported openly and interpreted correctly as a sharpness-versus-smoothness trade-off rather than hidden.

**Requested Changes:**

- Address statistical power directly. Three seeds are too few to support the canonical-fidelity claims, several of which fall within about one standard deviation (canonical BCE 0.0889 vs 0.0919; canonical Dice 0.8407 vs 0.8339). Either add more seeds (e.g., ten) so the small gaps can be assessed, or explicitly downgrade the canonical-BCE and canonical-Dice claims throughout the paper, including the abstract, to "comparable" or "weakly suggestive," reserving "improves" for the view-Dice and composition results that clearly exceed noise. Reporting per-seed values, and where feasible a paired statistical test, would strengthen this considerably.


- Adding a learned-warp condition would be an interesting experiment:
This model would keep the same inverse-sampling and bilinear warp structure as the rotor model, but instead of plugging in the known angle, it would learn the warp's rotation parameters from the pose code. It sits between the two current extremes: it has the spatial-warp machinery of the rotor model but the learned-from-data quality of the baseline. This would isolate whether the rotor model's benefit comes specifically from the action being analytic (the rotation is exact and fixed by the known angle) or merely from having a spatial-warp layer at all, a distinction that the current two-way comparison cannot resolve. It would sharpen the paper's central claim, but the existing ablation already supports the analytic-versus-learned conclusion well enough that this is a strengthening addition rather than a requirement.

---

> ### Author Response · Authors · 2026-07-06
> **Response to Reviewer bMWS**
>
> We thank the reviewer for the careful reading and for identifying the central claim accurately. This was very helpful because it allowed us to make the revision follow this interpretation more consistently.
>
> 1. [Requested change]: Address statistical power; three seeds are too few.
>
> We addressed this throughout the revised experiments. The main 2D comparison now uses ten paired seeds, and the training budget is increased from twelve to thirty two epochs. The subsection Exact known-pose actions improve view Dice and composition consistency in 2D reports the ten-seed mean and standard deviation. The appendix Per-Seed Values and Paired Tests reports compact per-seed values, paired CIs and paired t-tests.
>
> The revised main result is stronger and more carefully scoped. The GA rotor improves view Dice from 0.8406+-0.0166 to 0.9031+-0.0189 with a paired mean difference of 0.0625 and a 95% CI of [0.0484,0.0767]. It also reduces composition MSE from 0.0327+-0.0029 to 0.0104+-0.0007 with a paired oriented difference of 0.0222 and a 95% CI of [0.0202,0.0242].
>
> We also revised the wording around canonical metrics. The Abstract and the subsection Exact known-pose actions improve view Dice and composition consistency in 2D no longer claim a canonical reconstruction advantage for the analytic pathway. In the revised longer budget experiment, the coordinate decoder obtains the best canonical BCE and canonical Dice.
>
> 2. [Requested change]: Report per-seed values and paired statistical tests.
>
> We added the appendix Per-Seed Values and Paired Tests. It contains compact per-seed tables for the main 2D experiment, Rotated MNIST and the 3D voxel experiment. It also reports paired CIs and paired t-tests for the main 2D comparison. The text explains that the GA rotor has higher view Dice and lower composition MSE than the coordinate decoder in all ten main 2D seeds. On Rotated MNIST and 3D, the per-seed summaries support the more cautious interpretation, i.e. composition consistency is the most robust effect, while reconstruction metrics are more mixed.
>
> 3. [Requested change]: Add a learned-warp condition.
>
> We added this control in Controlled Problem and Model Family, the Main 2D comparison table, Rationale for Model and Dataset Choices, and the subsection Exact known-pose actions improve view Dice and composition consistency in 2D. The learned-warp model decodes canonical logits and uses a differentiable sampler, but the sampler rotation is predicted from pose coordinates rather than set to the known angle. This places it between the coordinate decoder and the exact-action pathway.
>
> This addition sharpens the central claim. The learned-warp control tests whether the benefit comes from having a warp layer at all or from applying the supplied pose exactly. In the main 2D experiment, the learned-warp control remains close to the coordinate decoder rather than to the exact-action rows. We therefore conclude that the key structural difference is the exact use of the supplied pose, not merely the existence of inverse resampling.
>
> 4. [Additional clarification]: Geometric Algebra versus equivalent matrix/STN warps.
>
> We strengthened this clarification in the Abstract, Introduction, Controlled Problem and Model Family, Exact known-pose actions improve view Dice and composition consistency in 2D, Discussion and Limitations, and Conclusion. The exact SO(2)/STN and GA rotor rows are expected to agree because they implement the same planar rotation through the same sampler. This is now presented as a transparency check.
>
> Because the reviewer also noted that the original paper only sketched the more interesting 3D setting, we added a controlled 3D voxel rotation experiment in the subsection A controlled 3D voxel experiment preserves the composition pattern. We present this as a rotation-only exact SO(3) action study, not as a full motor or SE(3) rigid motion experiment.
>
> The appendix Independent clifford Verification of 2D and 3D Rotor Calculations reports the algebraic conversion checks. We also clarify that these checks support the implementation of the exact action, but do not turn the 2D or 3D experiments into rotor versus matrix comparisons.

---

### Review · Reviewer_6WPx · 2026-06-24

**Summary Of Contributions:**

The paper studies whether pose-aware VAEs benefit from applying a pose explicitly via an exact transformation rather than learning to apply it implicitly by directly rendering the transformed object. To this end, the authors study a deliberately simplified setting of 2D binary shapes under planar rotation with known poses and closely matched architectures, ensuring that the main structural difference between the two models is the pathway through which the pose enters.
The encoder maps two rotated views of the same object to a single latent variable that is intended to encode the canonical shape of the object. The proposed model decodes this canonical shape and applies the rotation by a known angle $\theta$ explicitly, while the baseline concatenates $(\cos \theta, \sin \theta)$ to the encoding before feeding it to the decoder, which is tasked with reconstructing the rotated shape directly.

The authors evaluate the canonical and posed-view reconstructions (BCE and thresholded Dice) together with a relative-pose composition-consistency metric, averaged over three seeds in a compressed-capacity regime. The proposed model achieves improvements in view Dice and substantially reduces composition-consistency error.
The baseline attains a lower view BCE, since the implicit decoder can produce smoother boundary probabilities to lower BCE at the cost of degrading agreement of the binarized shape.

**Additional Comments:**

Have the authors considered a mixture-of-experts-style VAE [1, 2] to resolve the per-view vs. fused-posterior tension? I think such a formulation would admit a valid variational bound while naturally enforcing per-view content agreement, instead of relying on a fused reconstruction with per-view KL terms plus a separate invariance penalty.

[1] Yuge Shi, N. Siddharth, Brooks Paige, Philip H.S. Torr. Variational Mixture-of-Experts Autoencoders for Multi-Modal Deep Generative Models. NeurIPS 2019
[2] Thomas M. Sutter, Imant Daunhawer, Julia E. Vogt. Generalized Multimodal ELBO. ICLR 2021

**Audience:**

Yes

**Audience Explanation:**

I believe that while the general question the authors pose is interesting, in its current form, the value of this paper to the community is limited. As mentioned above, the findings seem unsurprising and cannot easily be extrapolated to more interesting settings where the geometric algebra does real work. For the presented 2D setting, the geometric algebra framing is purely decorative: it is necessary neither for the presented derivations nor for the implementation. Commendably, the authors clearly state these limitations, but the narrow scope leaves the paper without a clear takeaway other than largely re-confirming existing intuitions about geometric inductive biases. Adding the 3D setting, which is currently discussed as future work, would make the paper much stronger.

**Claims And Evidence:**

Yes

**Claims Explanation:**

The claims made in the paper are supported by the provided evidence, though the gains on the weaker metrics (canonical BCE and Dice) fall within the noise across seeds. While genuine, the tightly scoped contributions appears modest because the conclusions are unsurprising and provide little insight into more interesting settings where the geometric algebra does real work.

The paper compares the explicit application of a known transform against learning to apply it implicitly, which is interesting and distinct from the spatial-transformer comparison that concerns learning the transform itself. The reason the result is unsurprising is that building a known transformation directly into the architecture tends to beat learning to reproduce it. This is especially true given the small network capacity used in the experiments. If the decoder cannot exploit a shortcut by learning the application of the transformation implicitly rather than explicitly, which is hard to conceive in this setting, it is hard to imagine a different conclusion.
Moreover, the geometric algebra adds nothing over a standard rotation matrix in the presented setting, and several advantages hold entirely by construction, such as the content/pose separation (pose is supplied) and the composition-consistency property built into the analytic pathway (see Proposition 1).

**Requested Changes:**

- As mentioned above, I believe that adding the discussed 3D setting would significantly strengthen the paper
- Given that compute is not a limitation for the experiments, the reported results should be reported over more seeds.
- Given that the loss has 6 hand-set parameters and runs are cheap, an ablation of the different loss terms would be interesting and give more insight into how much each term contributes and how they interact with the different models. I think the confidence penalty, which influences the sharpness of the predicted logits and may amplify the thresholded metrics, and the invariance penalty, which influences content/pose separation, would be especially interesting.

---

> ### Author Response · Authors · 2026-07-06
> **Response to Reviewer 6WPx**
>
> We thank the reviewer for the constructive suggestions. We agree that the original contribution was narrow and that the paper needed more evidence on 3D, statistical robustness and the effect of individual loss terms. The review gave us the opportunity to substantially strengthen the manuscript while keeping the claim carefully scoped.
>
> 1. [Requested change]: Add the discussed 3D setting.
>
> We added a controlled 3D voxel experiment, introduced in Experimental Protocol and in the subsection titled A controlled 3D voxel experiment preserves the composition pattern. The experiment uses 32^3 binary occupancy volumes, known multi-axis rotations sampled from SO(3) and trilinear inverse sampling. The coordinate decoder receives a continuous 6D rotation representation, the learned-warp model predicts constrained rotation parameters before sampling and the exact-action model applies the ground-truth SO(3) matrix induced by the sampled unit quaternion/spatial rotor.
>
> We also clarified the role of Geometric Algebra in this experiment. The voxel sampler consumes affine coordinate maps, so the spatial rotor/unit quaternion is converted to its induced SO(3) matrix for sampling. The revised 3D results therefore report one exact SO(3) action row, not separate 3D rotor and matrix rows. The appendix 3D Rotations, Spatial Rotors and the Sampling Interface explains this relationship, and Independent clifford Verification of 2D and 3D Rotor Calculations verifies the algebraic conversion. The 3D result is not overstated, composition MSE drops from 0.0482+-0.0011 to 0.0202+-0.0008 with a modest view-Dice gain, while the coordinate decoder obtains the best canonical BCE, canonical Dice and sampled latent View BCE diagnostic.
>
> 2. [Requested change]: Report more seeds.
>
> We increased the main 2D comparison from three to ten paired seeds. The new Rotated MNIST and 3D voxel experiments also use ten paired seeds. The appendix Per-Seed Values and Paired Tests reports compact seed level summaries for the main 2D experiment, Rotated MNIST and the 3D voxel experiment, and also reports paired confidence intervals and paired t-tests for the main 2D comparison. This led us to sharpen the claim, i.e. we no longer claim a canonical reconstruction advantage for the analytic pathway. The revised positive claim concerns thresholded posed-view fidelity in the main 2D setting and consistently lower relative-pose composition MSE in the tested settings.
>
> 3. [Requested change]: Ablate the hand-set loss terms, especially confidence and invariance.
>
> We added the subsection Loss ablations show that the result is not driven by one hand-set term and the targeted loss-configuration ablation table. Each row retrains both decoder types under the same architecture, data split, seed set, optimiser and epoch budget while changing only the listed loss weights. Removing the confidence penalty leaves GA rotor view Dice at 0.9033±0.0316 and composition MSE at 0.0101+-0.0009. Removing the invariance term gives 0.9139+-0.0259 and 0.0104+-0.0009. The entropy column shows that the coordinate decoder often has lower entropy, so the rotor view-Dice advantage is not explained by sharper predictions alone. Metrics, Loss and Statistical Reporting defines the implemented loss terms.
>
> 4. [Requested change / comment]: Mixture-of-experts-style VAE.
>
> We added this point in Related Work and clarified it again in the model description and limitations. In particular, the revised text notes that mixture-of-experts, product-of-experts and generalized multimodal ELBO formulations are more principled multimodal variational objectives, while our shared latent averaging step is used only as a controlled fusion device to keep the pose-pathway comparison matched across models.
>
> 5. [Additional clarification]: GA appears decorative in 2D.
>
> We now state explicitly that planar rotors, SO(2) matrices and rotation-only spatial transformer warps are equivalent ways to implement the same 2D rotation. The 2D experiment is not presented as evidence that GA is numerically superior to matrices. We keep GA because it expresses the exact-action/composition viewpoint and extends naturally to spatial rotors for 3D rotations and to motors for future SE(3) rigid-motion experiments. The present 3D experiment is rotation-only, full SE(3) motor experiments are left for future work.
>
> 6. [Additional clarification]: Composition consistency is partly built into the analytic pathway.
>
> We agree and made this explicit. Experimental Protocol states that the composition metric checks whether outputs respect the supplied group action and is not an independent natural-image quality benchmark. The Results, Discussion and Limitations and Conclusion frame composition consistency as validation of the intended inductive bias, not proof of broad reconstruction superiority.